# Understanding Mn-nodule distribution and evaluation of related deep-sea mining impacts using AUV-based hydroacoustic and optical data

Anne Peukert[1], Timm Schoening[1], Evangelos Alevizos[1], Kevin Köser[1], Tom Kwasnitschka[1], and Jens Greinert[1,2]

[1]GEOMAR Helmholtz-Center for Ocean Research Kiel, Germany
[2]Christian-Albrechts University Kiel, Germany

*Correspondence to:* Jens Greinert (jgreinert@geomar.de)

**Abstract.** In this study ship- and AUV-based multibeam data from the German Mn-nodule license area in the Clarion-Clipperton Zone (CCZ; eastern Pacific) are linked to ground truth data from optical imaging. Photographs obtained by an AUV enable semi-quantitative assessments of nodule coverage at a spatial resolution in the range of meters. Together with high resolution AUV bathymetry this revealed a correlation of small-scale terrain variations (<5m horizontally, <1m vertically) with nodule coverage. In the presented data set, increased nodule coverage could be correlated with slopes >1.8° and concave terrain. On a more regional scale, factors such as the geological setting (existence of horst and graben structures, sediment thickness, outcropping basement) and influence of bottom currents seem to play an essential role for the spatial variation of nodule coverage and the related hard substrate habitat.

AUV imagery was also successfully employed to map the distribution of re-settled sediment following a disturbance and sediment cloud generation during a sampling deployment of an Epibenthic Sledge. Data from before and after the 'disturbance' allows a direct assessment of the impact. Automated image processing analyzed the nodule coverage at the seafloor, revealing nodule blanketing by resettling of suspended sediment within 16 hours after the disturbance. The visually detectable impact was spatially limited to a maximum of 100m distance from the disturbance track, downstream of the bottom water current. A correlation with high resolution AUV bathymetry reveals that the blanketing pattern varies in extent by tens of meters, strictly following the bathymetry, even in areas of only slightly undulating seafloor (<1m vertical change).

These results highlight the importance of detailed terrain knowledge when engaging in resource assessment studies for nodule abundance estimates and defining mineable areas. At the same time, it shows the importance of high resolution mapping for detailed benthic habitat studies that show a heterogeneity at scales of 10m to 100m. Terrain knowledge is also needed to determine the scale of the impact by seafloor sediment blanketing during mining operations.

# 1 Introduction

## 1.1 Nodule abundance estimation in relation to benthic structures

The deep ocean is an area of economic interest due to its potential reserve of metal resources. Before deep sea mining can be conducted, a better understanding is required of the ecological role of the deep sea as the largest habitat on earth. One focus lies on impacts of ferromanganese nodule (Mn-nodule) mining which recently has been studied in international projects like MIDAS (FP7 project 603418) and Mining Impact (JPI Oceans project). Mn-nodules form a hard substrate for sessile fauna (Purser et al., 2016; Vanreusel et al., 2016) and their removal is expected to impact respective fauna, but mobile fauna is impacted as well (Bluhm et al., 1995). Quantifying Mn-nodule occurrence and understanding distribution patterns on the seafloor is thus required for environmental baseline studies as well as ecological and impact assessments.

Several studies correlate bathymetry and nodule occurrence, revealing a complex/non-coherent interrelation which mainly depends on the considered spatial scale. Most studies have focused on nodule occurrence variability between very different terrain settings such as seamounts, valleys, plains and undulating terrain (Halbach, 1988; Pattan and Kodagali, 1988; Skornyakova and Murdmaa, 1992; Sharma and Kodagali, 1993; Park et al., 1997; Jung et al., 2001; Kim et al., 2012). The detected Mn-nodule variability has been associated with sediment deposition properties, e.g. assuming increased accumulation of sediment in flat or depression areas compared to sloping seafloor (Frazer and Fisk, 1981; Widmann et al., 2014). Local sediment accumulation influences the dominant formation type of the Mn-nodules (diagenetic vs. hydrogenetic), their size and metal concentration (Jung et al., 2001; Kim et al., 2012; Mewes et al., 2014; Widmann et al., 2014), but detailed small-scale investigations (1m to 100m scale) are not commonly done. (Okazaki et al., 2013) propose to perform such investigations to improve our knowledge about Mn-nodule formation processes and the affecting parameters. Moreover, the substrate changes considered in this study provide relevant information for estimating size and heterogeneity of local-scale habitats.

The study analyzes ship-based bathymetric data for large scale background information together with AUV-obtained high resolution multibeam (MB) and optical data to reveal detailed nodule coverage patterns within a 12km$^2$ area. The resolution of the acoustical AUV data enables the identification of vertical morphological undulations of less than 1m with a lateral resolution of 3-5m. Results from optical and hydroacoustic data were analyzed to assess correlations between Mn-nodule coverage and small-scale morphology and to extrapolate Mn-nodule occurrences to a wider area.

An equivalent approach was applied for an environmental impact study on sediment blanketing during a simulated 'mining-operation'. Mn-nodule mining will affect the seafloor and benthic fauna in several ways. A removal of the upper-most sediment layer (5-20cm) will cause habitat loss for sessile fauna which depends on nodules as hard substrate (Figure 1; (Vanreusel et al., 2016)), and for organisms living in the uppermost 'fluffy' sediment layer. The suspended sediment plume can clog filter organs of suspension feeders and the re-depositioning of suspended sediment will bury sessile organisms. These physical impacts could further be accompanied by bio-geochemical disequilibria in the water column and the sediment surface that will impact the local environment on short and long time scales (Shirayama and Fukushima, 1997; Kotlinski et al., 1998; Sharma et al., 2001; Thiel and Tiefsee-Umweltschutz, 2001). The release of toxic substances or metals might also add to the impact. Together these effects can increase mortality with unknown short term and cumulative effects (Markussen, 1994; Sharma, 2011). Increased

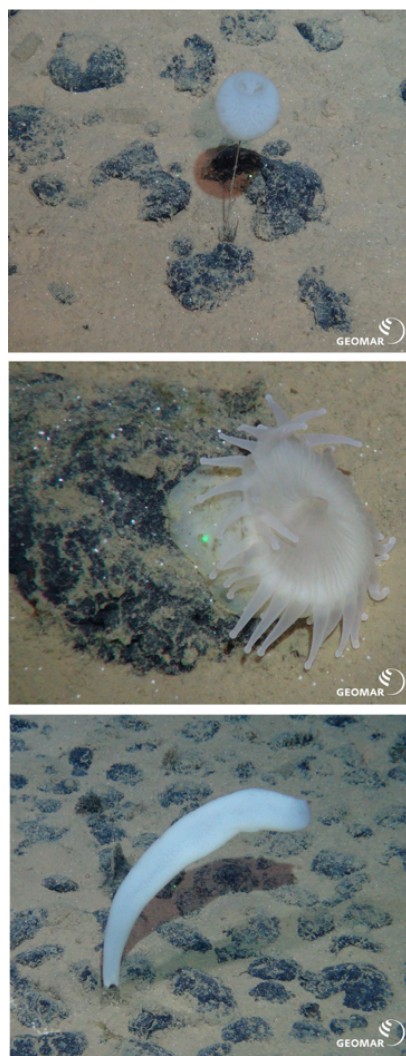

**Figure 1.** Sessile benthic organisms depending on manganese nodules as a hard substrate habitat. Images from the German claim area in the Clarion Clipperton Zone (photos: ROV Kiel 6000, GEOMAR-Helmholtz Center for Ocean Research Kiel, Germany)

water turbidity in the water column and the re-deposition of large volumes of re-suspended sediment in a relatively short time interval is expected during mining and are uncommon in the deep sea. Estimating the extent and distribution pattern of the re-settled particles of such sediment plumes is therefore relevant for assessing deep-sea mining impacts on a larger spatial scale.

## 1.2   Previous Benthic Impact Experiments and environmental studies

Various Benthic Impact Experiments (BIEs) have been conducted to study sediment re-suspension and the distribution of sediment plumes in Mn-nodule areas (e.g. (Ozturgut et al., 1980; Foell et al., 1990; Fukushima et al., 1995; Sharma, 2001)) for which data were acquired by (few) moorings equipped with different sensors, sediment sampling as well as by optical observations using video and photo material. These data finally contributed to the development of sediment plume distribution models (Lavelle et al., 1981; Jankowski et al., 1996) and led to interpolated blanketing maps (Barnett and Suzuki, 1997; Yamazaki et al., 1997). Observations from these larger-scale (as compared to thos study) BIEs (OMI, OMA, OMCO (1978), DISCOL (1989), BIE-II (1993), JET (1994), IOM-BIE (1995), INDEX (1997), MMAJ (1997)) indicated different distribution distances of the created bottom plume ranging from several tens or hundreds of meters (Barnett and Suzuki, 1997; Trueblood et al., 1997; Sharma, 2001) up to several kilometers away from the disturbances (Burns, 1980; Lavelle et al., 1981; Jankowski et al., 1996; Yamazaki et al., 1999; Yamada et al., 1998).

The use of different disturbance gear for different duration per BIE leads to inconsistent interpretations (Jones et al., 2000). Unfortunately, definitions for 'a sediment plume' differ with regards to minimum particle size and amount of particles and thus, parameters are applied differently in model approaches. While some models calculate the distribution of the re-suspended material until all sediment particles have settled from the water column (pers. comm. A. Dale, SAMS), others define certain particle concentration thresholds (Burns 1980). A concise plume definition could be based on thresholds (e.g. with regard to amount of particles, shape and size distribution) that benthic organisms could tolerate on short but also longer cumulative time scales. Defining such thresholds requires in-situ experimental data of the reaction of benthic fauna to sediment plumes in the area of potential mining. Detailed studies do not exist but are essential for estimating the ecological consequences of deep sea mining.

Plume model results are based on several assumptions to include parameters describing the environment. Particle sizes and settling velocities are key factors in modelling plume distribution distances (Jankowski et al., 1996) and uncertainties can lead to miss-interpretations. Our current understanding of the behavior of re-suspended particles in the deep sea is based on laboratory experiments. Often, these experiments struggle to correctly determine settling velocities of flocculating particles, and/or they rely on specific deep sea sediments and might not account for the correct environmental parameters. Nevertheless, modeling the distribution of a plume induced by Mn-nodule mining is the only way to predict the possibly impacted area. Meaningful models need to incorporate all environmental aspects and need to operate at the highest possible resolution.

The study presented here focuses on an area within which the resettled sediment was visually observable in deep sea photographs of the seafloor. Two AUV photo surveys over the same area were conducted before and after the deployment of an Epibenthic Sledge (EBS), that created a sediment plume. The two data sets are directly compared to determine the scale of the visible disturbance. The correlation of the photo data with AUV-obtained bathymetry data reveals the influence of the local terrain variability on the sediment blanketing pattern and thus the sediment plume spreading.

## 1.3    Study area description

The study area is part of the Prospective Area 1 (PA1) within the German license area in the eastern Clarion Clipperton Zone (CCZ). The wider area (Figure 2) has a mean water depth of 4240m with abundant isolated or chains of seamounts of various heights. Mostly N-S-trending parallel graben and horst structures originating from the East Pacific Rise ((Rühlemann et al., 2011), Figure 2) can be seen as well. The PA1 itself is a plateau-like area, elevated approximately 150m above the surrounding terrain and classified as a 'mineable plateau' in Figure 2D. Small scale undulations (<100m) on the plateau are smooth with slopes of <10° in wide parts of the area. A stronger relief is associated with seamounts or ridge structures (Figure 2). To the West, the plateau is bounded by a deep graben structure, whereas towards the East and the South the terrain slopes down to depressions with irregularly scattered seamounts of different sizes (red areas in Figure 2D).

High resolution studies using data from several AUV deployments were carried out within the 'mineable plateau' (black square in Figure 2). This plateau is characterized by slopes of less than 3° and the area is considered suitable for Mn-nodule collector systems (Kuhn et al., 2011). The AUV study area is located west of a ridge that follows the characteristic N-S lineation and is only little structured showing a smoothly undulating terrain (Figure 3). Towards the North the AUV study area is bounded by a slightly elevated area, which dips towards the South, leading into a wider depression. Smaller basins/depressions are found throughout the entire area. Slopes are generally <10° and those exceeding this value are found only in association with a group of seamounts in the South-West (Figure 3). The zoomed-in view in Figure 3 is based on newly acquired EM122 data (Greinert, 2016) showing a terrain more variable as initially assumed from the data set shown in Figure 2 (data from an EM120, 2° by 2° beam angle system). The more variable data also point towards a further terrain differentiation and potentially a less homogenous Mn-nodule coverage within the area. The acquired AUV multibeam data that were processed to a horizontal resolution of 5m provide the small-scale morphology. The resulting bathymetric maps were used for correlating nodule coverage and sediment blanketing patterns of the disturbance experiment.

## 2    Methodology and Disturbance Experiment

All ship- and AUV-based surveys were conducted in March 2015 during the 'EcoResponse' cruise SO239 with R/V SONNE (Martínez Arbizu and Haeckel, 2015). Large-scale bathymetric data were acquired by the hull-mounted Kongsberg EM 122 Multibeam Echosound (MBES) system (12kHz, 0.5° along- and 1° across-track beam angle; 55m cell size). The swath angle was set to 120° and the survey speed was about 8kn. The REMUS 6000 type AUV Abyss was deployed for the high resolution mapping and photo surveys (Linke and Lackschewitz, 2016). A RESON Seabat 7125 MBES system was used (200kHz, 2° along track and 1° across track beam angle). The AUV-based multibeam surveys were conducted at an altitude of 80m above the seafloor. Bathymetric maps were produced with the software packages GMT 5.2 (Wessel et al., 2013) and ArcGIS 10.2. The MBES data were analyzed in ArcGIS 10.2 as a floating point raster (see Appendix for cell size of different regions, Table A1). Data were projected as UTM coordinate system (Zone 11N) to enable spatial analysis. Bathymetric first order derivatives (slope, aspect) and second order derivatives (Bathymetric Position Index - BPI, Vector Ruggedness Measure - VRM, total curvature, plan curvature, profile curvature) were calculated for each region and sub-region using tools in SAGA GIS and

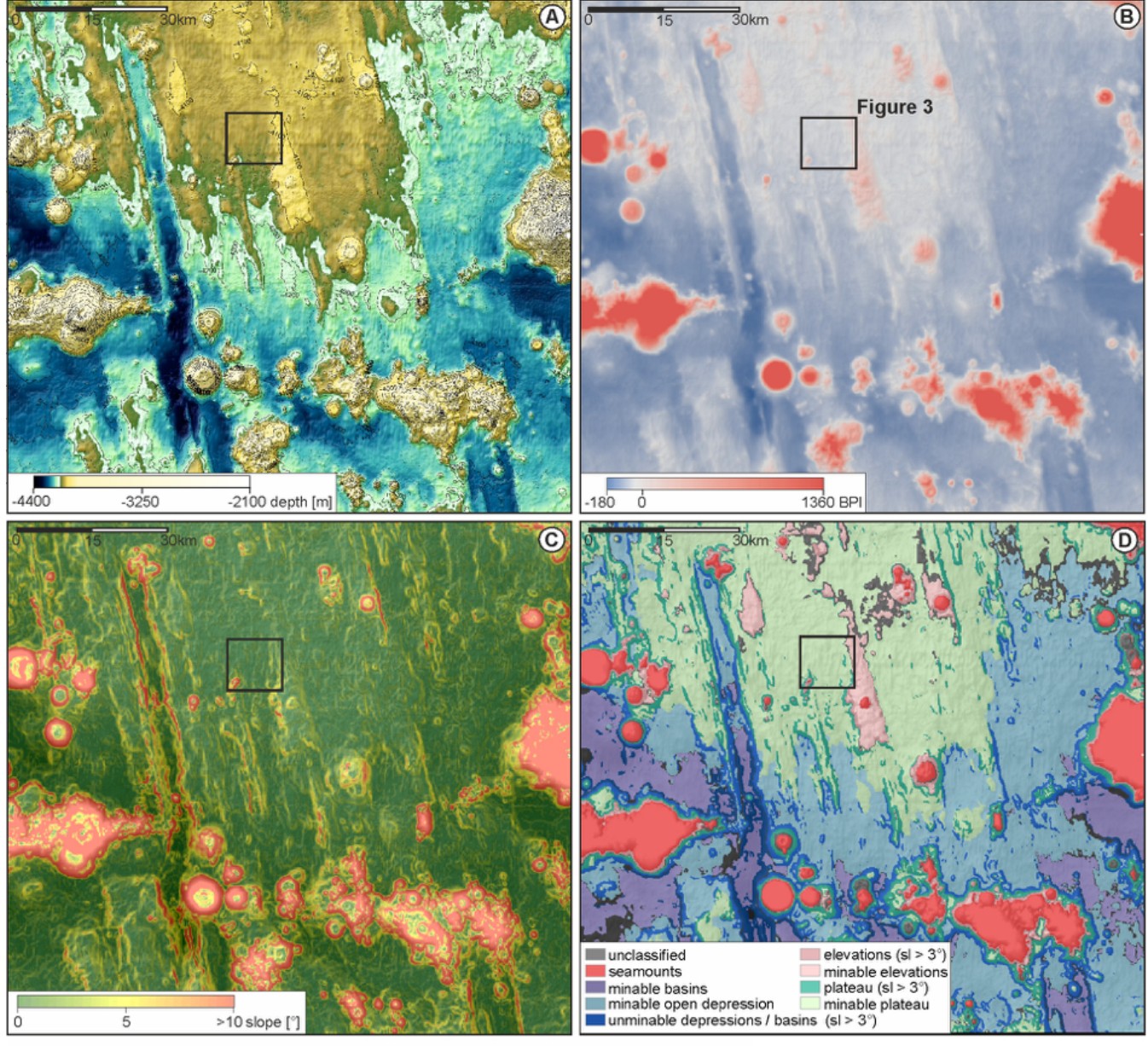

**Figure 2.** Overview maps of PA 1 showing A: Bathymetry, B: Bottom Positioning Index (BPI) with scale factor 11,000 (grid cell size: 110m, inner radius: 10 cells, outer radius: 100 cells), C: Slope and D: Classification of the terrain based on the classification dictionary in Appendix Table A2 (sl = slope). The terms *mineable* and *unmineable* are defined by the slope threshold (here *mineable* = slope $\leq 3°$; *unmineable* = slope $> 3°$; this is rather conservative threshold, current discussions mention $7°$ as more realistic). Black squares mark the study area (center $117°1W\ 11°51N$) shown in Figure 3.

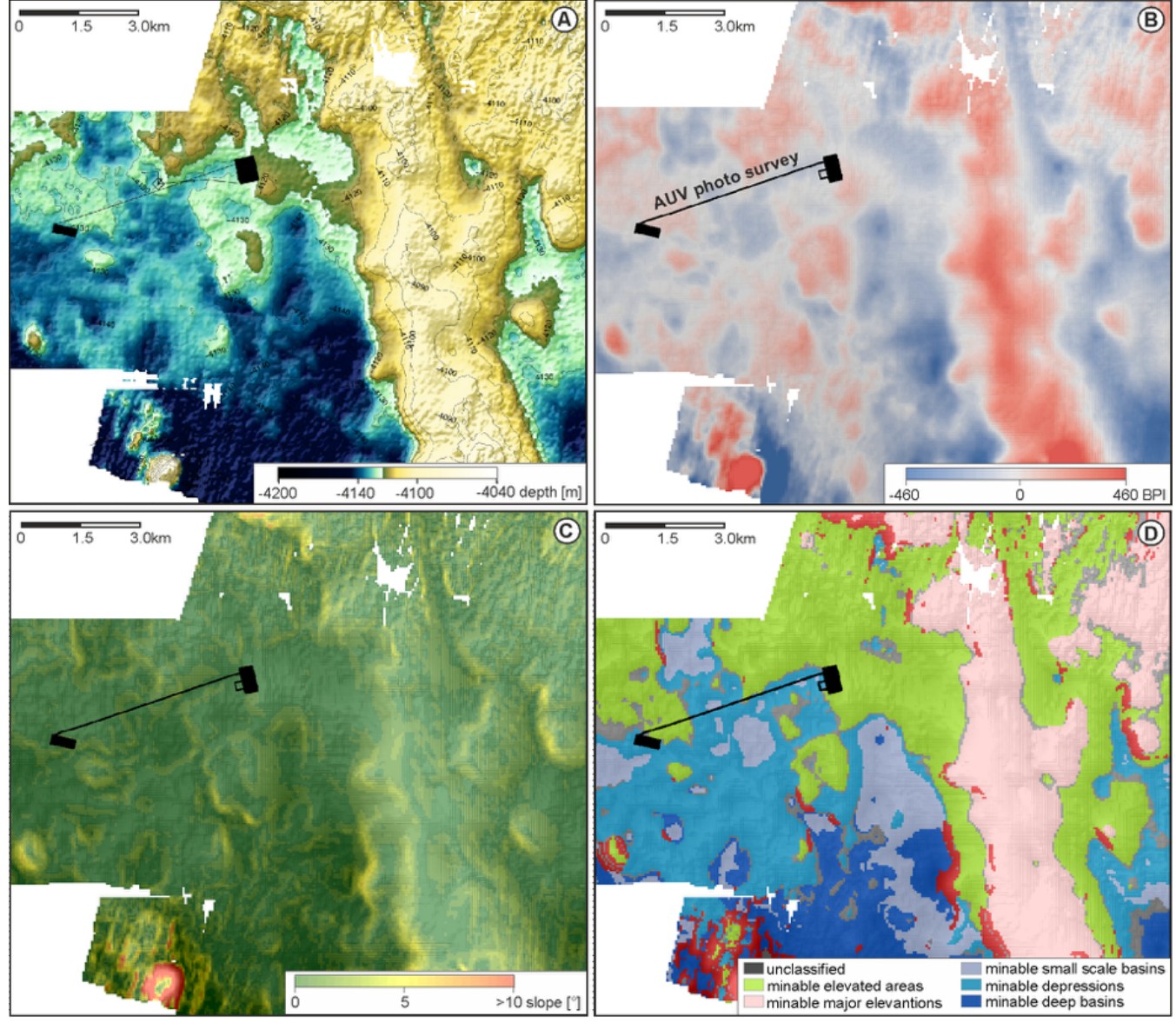

**Figure 3.** Overview maps for the geological setting of the AUV study area. A: Bathymetry; B: Bottom Positioning Index (BPI) with scale factor 2,750 (cell size: 55m, inner radius: 10 cells, outer radius: 50 cells); C: Slope (in degree); D: Terrain classification based on the classification dictionary in Appendix Table A3. Black lines indicate the track of AUV Dive 168 prior the EBS deployment (A) and the locations of usable photos from this dive (B, C and D). See Figure 2 caption for the definition of "mineable" area.

ArcGIS (spatial analyst toolbox and the 'Benthic Terrain Modeler' Add-on toolbox (Wright et al., 2012); for further details see Appendix and Table A4).

The AUV camera system 'DeepSurveyCam' (Kwasnitschka et al., 2016) was used for visual seafloor inspection by two photo surveys. Photos were taken from 7 to 9m altitude at a mean speed of 3kn, gaining more than 50,000 usable photos. Two sub-areas A1 and A2, located approximately 5km apart, were photographed extensively. One part of both AUV-surveys in sub-area A2 followed exactly the same track before and approximately 16h after the deployment of an Epibenthic sledge (Greinert et al., 2017; Schoening, 2017a, b). Photos were automatically analyzed for the Mn-nodule coverage on the seafloor (percent coverage) and nodule size distribution (different quantiles in $cm^2$, see Appendix for details). The 'Compact Morphology Nodule Delineation' algorithm was used for this task (Schoening et al., 2017). This nodule delineation method is based on color differences between the nodules and the sediment background. Derived quantitative values are georeferenced and can thus be jointly analyzed with the AUV-obtained bathymetry. This allows understanding correlations between nodule abundance and the decameter-scale morphological changes in the AUV studied area.

As a side product of benthic sampling using a B-EBS Type sledge (Brenke, 2005) a sediment plume was created. The sledge itself has a length of 360cm and a width of 120cm with a weight of approximately 420kg in water. It creates a pressure of about $13g/cm^2$ onto the sediment surface (see Appendix Figure A3). The sledge was towed during station SO239_024-EBS across sub-area A2 (Figure 4A) from West to East at ca. 0.5m/s leaving a track of approximately 20cm in depth (Martínez Arbizu and Haeckel, 2015). The re-settlement of the sediment plume was visually mapped using the camera system and automated image analysis mentioned above.

## 3 Results

### 3.1 AUV-based bathymetry and overview of Mn-nodule coverage

The water depth within the AUV-mapped area ranges from 4110m to 4143m (Figure 4A) with 93% of the area showing slopes <3° (Figure 4B). Steeper slopes between 7° and 10° occur locally in the East and towards the North as well as in randomly distributed pit structures, which occur throughout the area (Figure 4). These pit structures take up approximately 10% of the area and attain sizes from several tens of meters to 150m in diameter with a maximum depth of 4m. They occur exclusively within larger depressions (ca. 50% of the area) as visualized by the BPI map (Figure 4C).

The two AUV photo surveys provide visual data from within the high resolution MBES map (black lines, Figure 4A) covering a depth range from 4134m to 4114m. The first AUV camera survey (SO239_019_Abyss168) provided data over two extensive sub-areas in the West (A1) and East (A2), as well as one survey line connecting both areas (Figures 4A & 5). The second survey remapped parts of sub-area A2 (SO239_028_Abyss169) and additionally mapped a similarly sized area further south (Martínez Arbizu and Haeckel, 2015). Based on the automated image analyses the majority of the seafloor shows nodule coverage values between 8% and 17% (Figure 5A). Values below and above this range (<1%) have been neglected, since they are caused by "unusual" objects (like EBS-tracks or organisms) in the images.

In the following examinations the threshold between 'low' and 'high' Mn-nodule coverage is set at 12.5%, which is the analyzed mean coverage value of the considered range. In the eastern A2 sub-area a greater proportion of higher coverage values (13-16%) can be observed. A positive correlation was found between Mn-nodule coverage and median size of the nodules

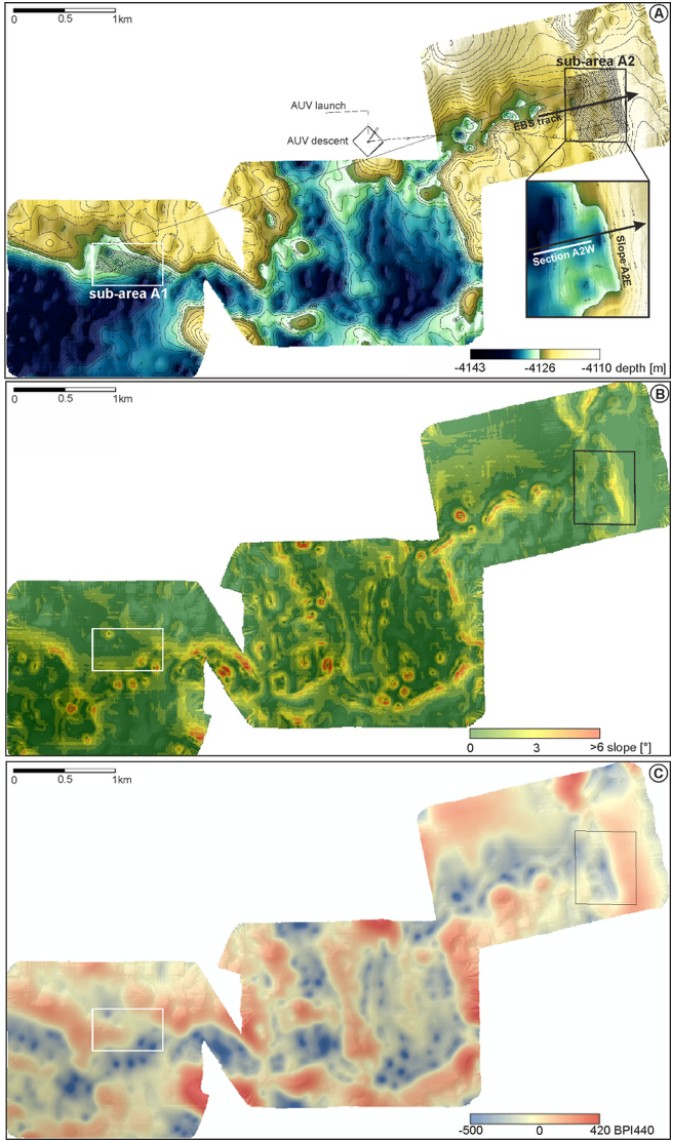

**Figure 4.** Bathymetric map obtained by the AUV with black line indicating the track of AUV Abyss Dive 168 prior to the EBS deployment. The black arrow marks the tow track of the EBS deployment. B): Slope map derived from bathymetry indicating maximum slopes of $10°$; C): BPI map (BPI440) derived from AUV-obtained bathymetry. The black and white rectangles indicate the Eastern and Western sub-areas.

(Figure 6A). The correlation decreases with increasing nodule size, indicating a compensation of the size by a decreasing number of occurrences. This is consistent with findings of former studies e.g. by (Okazaki et al., 2013).

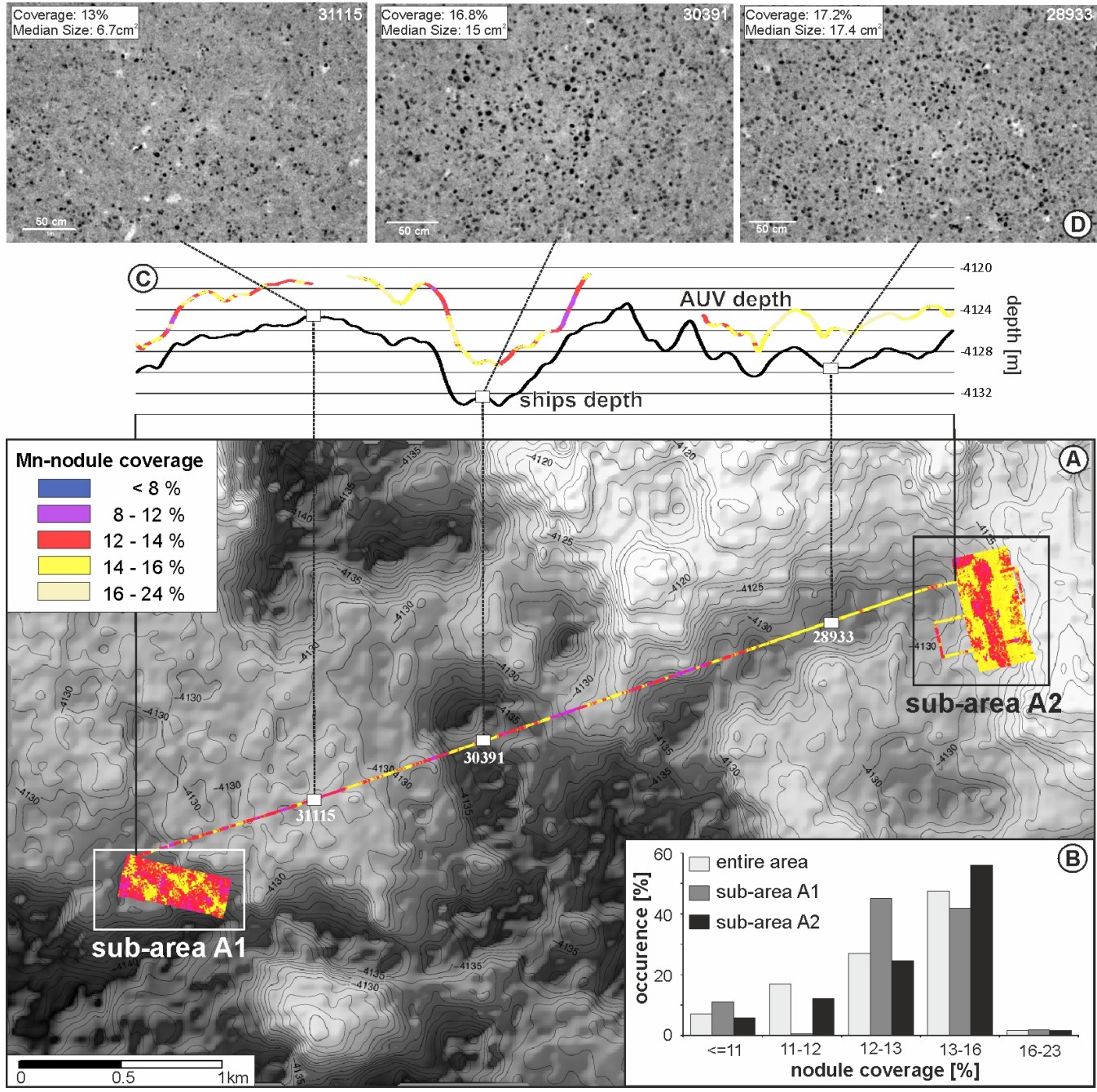

**Figure 5.** A) Bathymetric map (ship-based MBES) of the Working Area with nodule coverage calculated from AUV photo survey; the photo examples in D) show the Mn-nodule coverage and median size in different areas. B) Statistical evaluation of nodule coverage for the entire photo survey (green bars; 30,038 photos) and the sub-areas A1 (gray bars; 10,120 photos) and A2 (red bars; 16,890 photos).

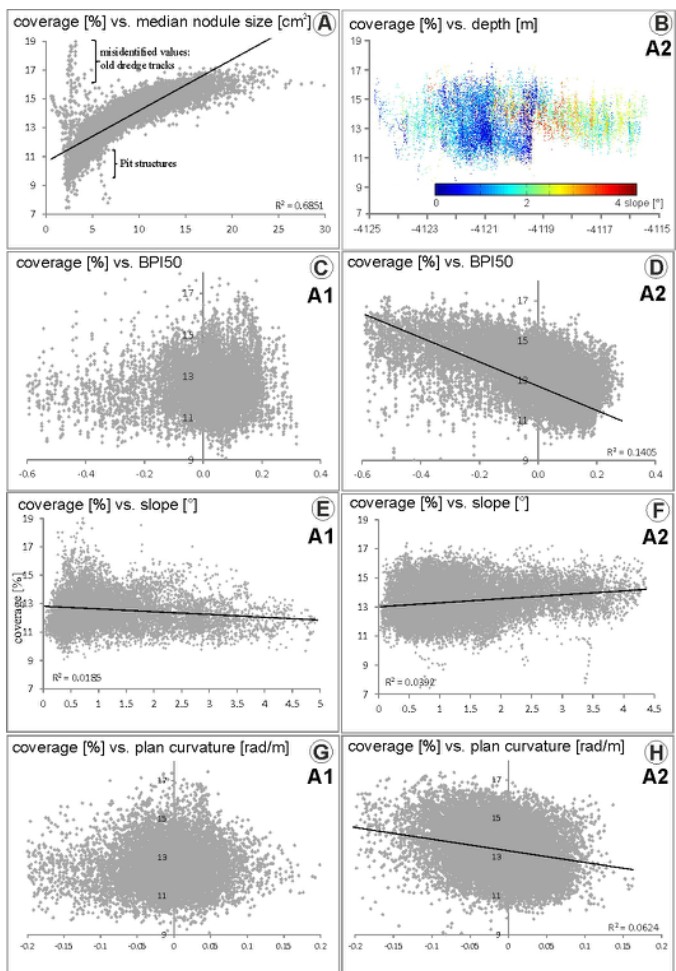

**Figure 6.** Scatter Plots indicating relationships between Mn-nodule percent coverage (%) and eight other nodule and terrain values: median nodule size in cm$^2$ (A), depth within the A2 sub-area (B, color-coded by slope), BPI50 (C,D), slope (E,F) and plan curvature (G,H). Charts (C,E,G) relate to sub-area A1; charts (B,D,F,H) to sub-area A2. The sub-areas show different correlations.

### 3.1.1 Broad scale variability (less detailed, correlation with ship-based bathymetric data, resolution 100-1000m)

Parts of dive SO239_019_AUV2 run across the entire working area providing data from different terrains that can be linked to the ship-based bathymetric information. The correlation between photo analysis and this less resolving bathymetry indicates a trend of decreasing nodule coverage at elevations and steeper sloping areas (Figure 5). Video-data acquired during previous cruises provide similar observations (Kuhn, 2015). The distribution pattern seen in the imagery also points towards small scale Mn-nodule coverage variability which is possibly related to minor topographic changes in meter to sub-meter scale. As only one track covers the central region of the working area clear correlations between Mn-nodule occurrence and large scale ship-based bathymetry are difficult to assess. Finding clear correlations is further complicated by the uncertainty of the AUV

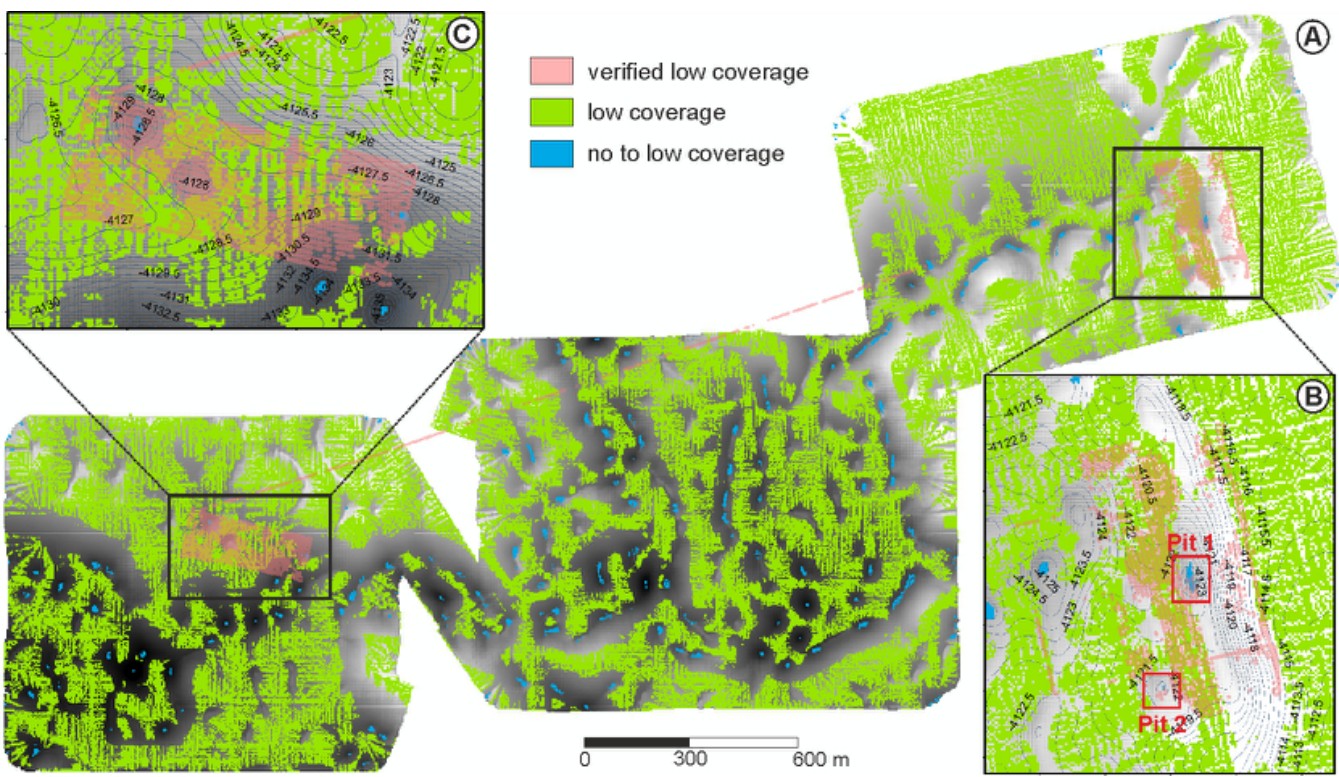

**Figure 7.** (A): Depth-shaded area, mapped hydro-acoustically by the AUV; (B): A2 sub-area, (C): A1 sub-area. The green shading indicates low nodule coverage areas (<=12.5%) that have been classified based on the correspondence in sub-area A2 (low nodule coverage corresponds with slopes <= 1.8° and positive plan curvature). Red boxes in B mark distinct, almost nodule-free, morphological depressions (Pit 1 & 2). Blue shades indicate areas with a high probability of very low to no Mn-nodule coverage. See classification dictionary in Table A5.

navigation (up to 30m), which prevents a precise geo-referencing of the photos between sub-area A1 and A2. More robust visual reference data could be provided by conducting a sparse mesh survey across the entire area or by a contiguous photo mosaic across different terrains.

### 3.1.2 Local scale variability (more detailed, correlation with AUV-based bathymetric data, resolution 1-100m)

The assessment of small scale Mn-nodule coverage heterogeneity was based on the western A1 and eastern A2 sub-areas; here, overlapping photo-mosaics and AUV-based bathymetric data in meter resolution exist (Figure 4A). Sub-area A2 (700m x 500m, 0.35km$^2$) is bound to the East by a 5-7m high 'ridge' with a relatively steep slope ('Slope A2E', 3°-7°). The western part of this area ('A2W') shows only minor morphological variation and a total relief of ca. 2m (Figure 4). Despite the rather small relief changes, variations in Mn-nodule coverage can be observed (Figures 5 & 7).

Figure 7A illustrates the detailed bathymetry of the studied area with red dots indicating lower Mn-nodule coverage ($\leq$12.5%) as indicted by image inspection. Those areas with a BPI50 > 0, slopes $\leq$ 1.8° and plan curvature values > -0.02 radians/m were

found to show the best correlation with the lower Mn-nodule coverage in sub-area A2 (Figure 6D, F, H). A NW-SE oriented, elongated patch in the central part of A2 that corresponds to a flat-topped (slope ≤1.5°), slightly convex shaped elevated structure (<1m above the surrounding terrain) shows a low Mn-nodule coverage. A higher Mn-nodule coverage instead occurs at

steeper slopes (>1.5°) and in morphological depressions (negative BPI values, negative plan and total curvature) indicating a sediment depositional environment. Two distinct depression structures (Pit 1 and Pit 2 structures in Figure 7), both approximately 60-80m in diameter and 1-2m deep, show a different pattern; here, the visible Mn-nodule coverage is significantly lower (0-8%). The almost spherical pit structures are bound by slopes of >2° and thus produce slightly increased Vector Ruggedness Measurement values >1x10-4 (VRM; Figure A4 A) and the lowest observed BPI-values (Figures 4C & A3 B). Similar struc-

tures are observed throughout the entire study area (blue shaded areas in Figure 7 and circular features seen in the slope map of Figure 4B). Based on additional ROV and benthic camera surveys it is assumed that these pit structures exhibit very few to no Mn-nodules Peukert (2016).

No further correlation between Mn-nodule coverage and bathymetric derivatives was found and no relation to absolute water depth could be observed (Figure 6B). However, Figure 6B shows a significantly lower variability of Mn-nodule coverage

for water depths shallower than 4019m (only ca. 4% variability, compared to 6-7% variability in deeper areas); these areas correspond to steeper slopes (ca. 2° to 3.5°) associated with the eastern bounding elevation ('Slope A2E', Figure 4A) of sub-area A2. Along this west-facing slope the Mn-nodule coverage clearly increases with increasing depth. Areas featuring low slope values show higher variability in Mn-nodule coverage (Figure 6B).

A lower Mn-nodule coverage (<12.5%) is predicted for the green areas marked in Figure 7A when using the BPI50, slope

and plan curvature classification of the A2 sub-area (Figure 7B). Although the resulting area does not match completely with the areas of low coverage derived from the photo analyses (red dots/shades in Figure 7), it represents the best correlation that could be achieved. Based on this result, a Mn-nodule coverage of <12.5% can be expected in 39% of the study area (green shades in Figure 7) and is likely to be very low or zero in at least 1% of the area (blue shaded parts).

In sub-area A1 (230 x 600m, 0.138km$^2$) no correlation is observed between the photo analyzed Mn-nodule coverage (red)

and the seafloor classification of A2 (green; Figure 7C). In addition, scatterplots (Figure 6C-H) show different dependencies between Mn-nodule coverage to BPI, slope and plan curvature between A1 and A2. In both areas though, the coverage attains more uniform values towards steeper slopes (Figure 7 E, F). A stronger correlation is shown in A2, and an inverse correlation in A1. In both areas the highest variability but also the lowest values of Mn-nodule coverage occur in generally flat areas (curvature values around 0, low slope values; Figure 7E, G).

Comparing the terrain statistics of areas A1 and A2 (Figure 8) reveals differences in their bathymetric settings, which might cause this discrepancy. Sub-area A2 mainly slopes towards West to Southwest, as indicated by the aspect distribution and to a lesser degree in 'opposite' Northeast to East directions caused by the general pattern of N-S striking graben and horst structures. In comparison, the main slope direction in A1 is towards a southerly direction. The slope distribution in A1 indicates a dominance of slopes up to one degree. This is reflected by the large area of flat seafloor as determined by the AUV-

BPI440 value distribution (Figure 8). In A2 slopes are steeper, the terrain is more variable and a larger number of depressions are observed compared to A1. The VRM shows similar values for both areas (Table A6). These differences in bathymetric

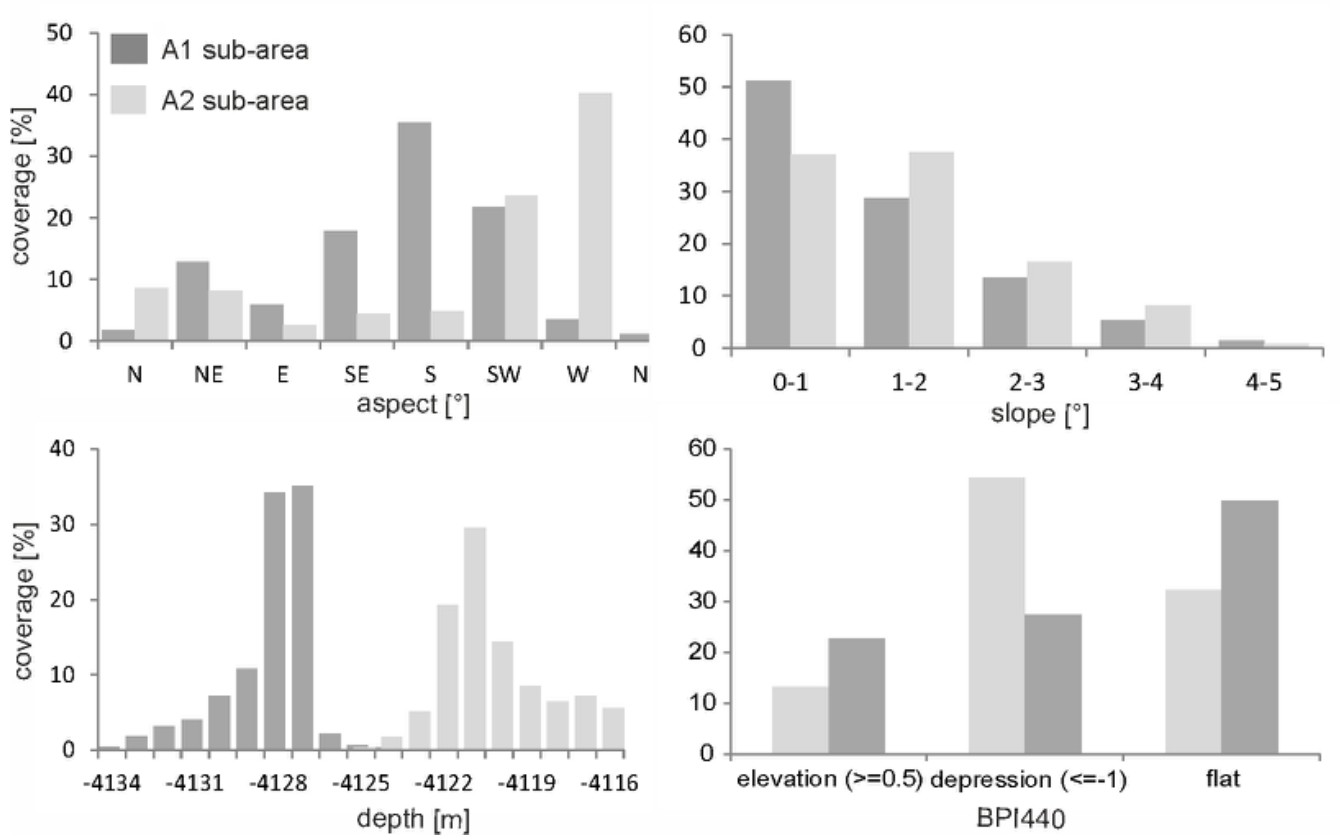

**Figure 8.** Distribution of derivative values in sub-area A1 and A2.

derivative values point at a lower terrain variability in A1, confirmed by the more consistent depth values in A1 relative to A2. Considering the generally lower Mn-nodule coverage within A1 (Figure 5A) it is concluded that lower Mn-nodule coverage correlates with lower terrain variability and lower slope values. This generalized observation is consistent with findings for A2. Although a direct one-to-one relationship valid in both sub-areas could not be derived, the general trend indicates higher Mn-nodule coverage with more variable terrain, along smooth slopes and in concave shaped terrain (depressions).

### 3.2 Sediment plume re-settling

To evaluate sediment plume re-settling, results of the automated image-based Mn-nodule detection before the EBS disturbance (SO239_019_Abyss168 with 6,061 usable photos) and after the EBS disturbance (SO239_028_Abyss169 with 10,783 usable photos) were compared (Figure 9A). Areas with the lowest analyzed coverage were associated with sediment blanketing that covers the Mn-nodules completely (here defined as <8% Mn-nodule coverage, green shaded areas in Figure 9).

The AUV-tracks of the photo surveys run perpendicular to the EBS track. A strong sediment blanketing can be observed close to the disturbance track (Figures 9 & 10). The photo mosaic shows a sharp transition between low (no) and higher

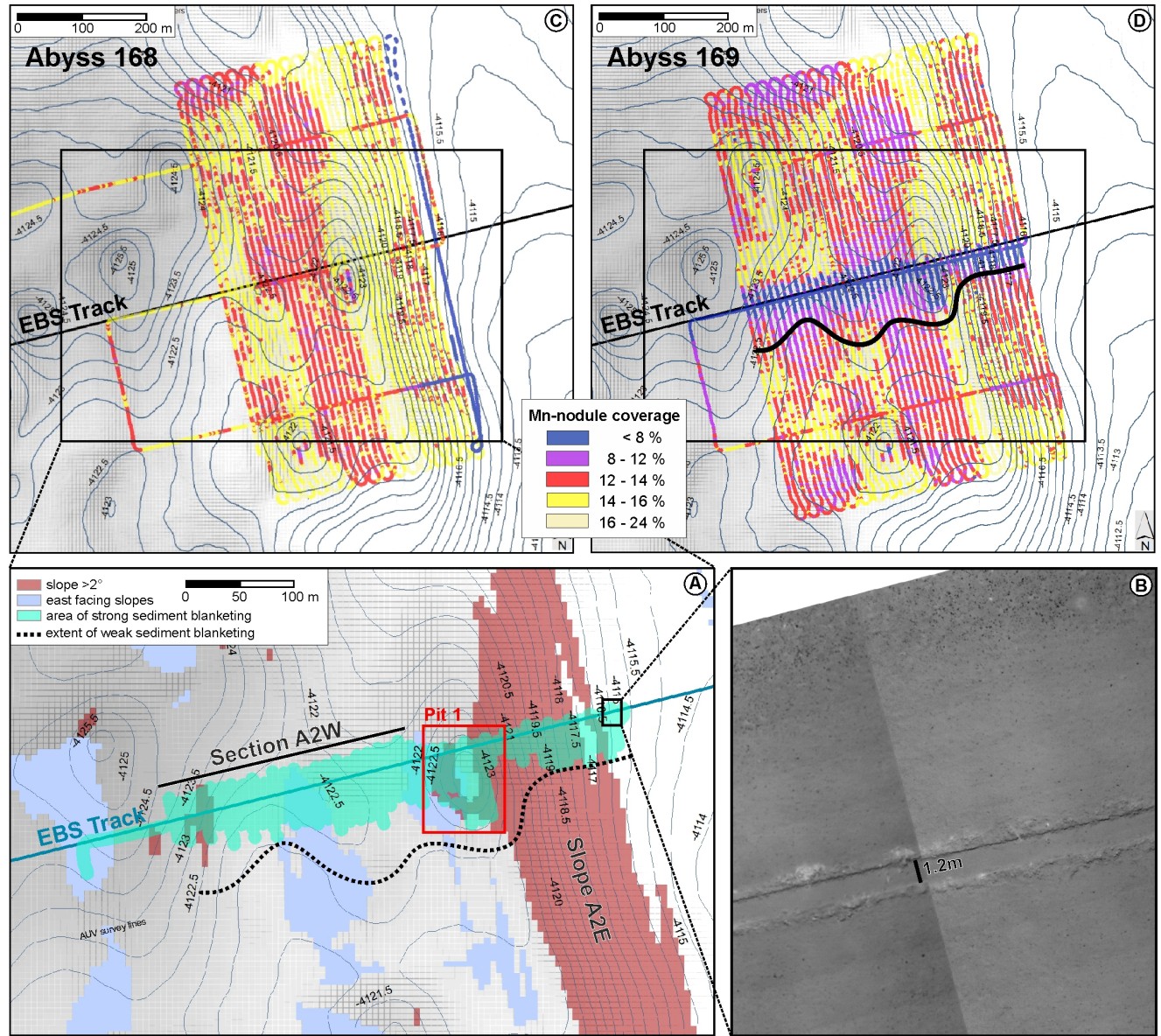

**Figure 9.** (A) The green shaded area in the detailed bathymetric map marks lowest nodule coverage sites, associated with the sediment cloud dispersal. The dashed black line represents the furthest extent of weak nodule blanketing. It corresponds to a nodule pixel color brightening and can be automatically computed from the imagery. Purple shading indicates east- facing slopes. Red shade marks areas with slope >2°. The blue line marks the ideal track of the EBS deployment. (B): Extract from a photo mosaic created from AUV imagery; nodules are completely covered by sediment towards the sides of the tracks; approximately 10m north of the track the sediment cover disappears within a sharp transition. Maps in (C) and (D) show the analysed Mn-nodule coverage before (C; dive SO239_019_AUV2) and after (D; dive SO239_028_AUV3) the EBS deployment.

Mn-nodule coverage north of the EBS track over a distance of 5m to 20m (Figure 10). South of the track the transition from complete sediment blanketing to areas without visible sediment cover is gradually fading out with increasing distance to the EBS-track; a slight sediment blanketing of Mn-nodules can be observed up to 70m away from the EBS track (Figures 9 & 10).

This pattern indicates a southward directed bottom current, which is confirmed by ADCP-based current measurements (station SO239_005; see Appendix Figure A5). An upward looking 300kHz ADCP (15min ensembles, 2m bin sizes) was positioned 500m SE of the EBS track at the time of the EBS deployment (Martínez Arbizu and Haeckel, 2015).

The combination of AUV-obtained bathymetry and imagery reveals a distinct blanketing pattern depending on the small scale morphology (Figure 9). In section A2W, where the total relief is only 1-2m, the re-suspended material was distributed

20m to 30m towards the north of the EBS track and 40m to 50m towards the south of it. West of A2W, within Pit 1 and the adjacent slope area A2E (Figure 9, red shaded area) the seafloor slope increases to max. 6° and the water depth decreases by several meters (Figure 9). This morphological change causes the sediment plume to cover the seafloor only up to 6m towards the north of the EBS track. The greatest distance at which sediment has been deposited away from the EBS track occurs within Pit 1 (Figure 9). Here the least visible Mn-nodule coverage extends up to 70m south of the EBS track, significantly further than

in the flat section A2W and slope area A2E.

## 4  Discussion

### 4.1  Uncertainties of photograph-based Mn-nodule coverage and size estimates

Seafloor photographs have been used for Mn-nodule occurrence studies for almost two decades (Park et al., 1997; Sharma et al., 2010; Okazaki et al., 2013). The presented AUV imagery reveals the natural Mn-nodule heterogeneity at the seafloor

surface on a very fine scale (decimeters) over an extensive area of 0.49km$^2$ that is completely photo-mapped. This highly detailed insight is of importance for a spatially detailed evaluation of the small scale habitat distribution and potentially allows a better resource assessment. However, potential uncertainties for the absolute numbers of Mn-nodule size and coverage are explained here.

Photographs only provide information of the sediment surface and thus will not be able to detect buried/sediment-covered

Mn-nodules (Sharma and Kodagali, 1993; Sharma et al., 2010, 2013), resulting in a potential underestimation of the absolute Mn-nodule abundance (Kuhn and Rathke, 2017). For absolutely accurate resource assessments and verification of the presented results, detailed sampling based on this study would need to follow. With respect to mapping Mn-nodules as hard grounds for sessile fauna, photographs give a realistic quantitative representation of size/coverage and spatial nodule abundance changes. Nevertheless, the automated image analysis CoMoNoD has some uncertainties (Schoening et al., 2017). In general, the nodule

identification employed here is based on contrast differences between the nodules (dark) and the sediment (bright). Mn-nodules on the seafloor could be located too close to each other to be correctly separated by the applied algorithm and, depending on the image quality and the contrast thresholds, quantitative coverage and size distribution values can be inaccurate (Sharma et al., 2010; Schoening et al., 2012; Tsune et al., 2014; Schoening et al., 2016, 2017) (Figure A6). Turbidity in the water, backscatter from particles, the water properties and the altitude of the camera (AUV) impact image quality ((Edwards et al.,

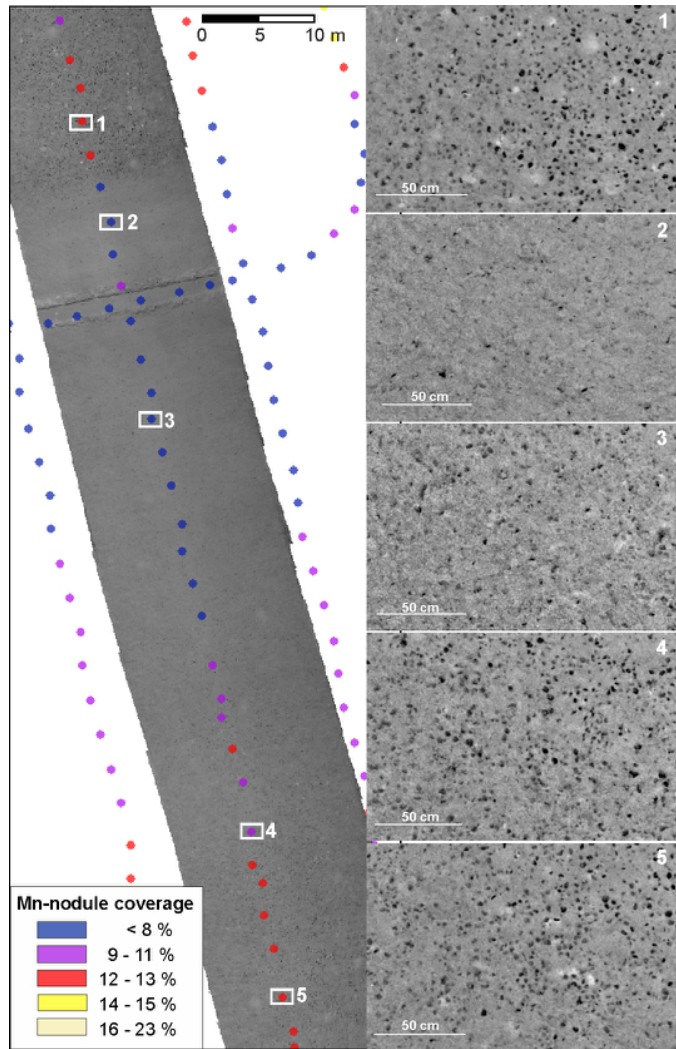

**Figure 10.** Section of the photo mosaic along one survey track line, with calculated nodule coverage values, indicated by the color coded dots (representative for the center of each individual photo which are ca. 15m x 11.5m (172m$^2$) in size). White squares mark positions of the enlarged photos shown to the right.

2003; Kwasnitschka et al., 2016). Nevertheless, optical imaging provides information on relative changes in seafloor Mn-nodule coverage and nodule sizes. This makes the presented technique an effective monitoring tool for habitat and environmental impact assessments that investigate the re-settling of suspended sediment. We detected a distinct trend of higher coverages correlating with larger nodules (Figure 7A) that could be a result of imperfect segmentation if the nodule density is too high.

## 4.2 Correlation between bathymetry and nodule occurrence

### 4.2.1 Regional scale correlation between ship-based bathymetry and Mn-nodule coverage/size

In general, properties such as sedimentation rate (Frazer and Fisk, 1981; Mewes et al., 2014), type and thickness of the sediment (Frazer and Fisk, 1981; Jeong et al., 1994) are believed to determine Mn-nodule growth (von Stackelberg and Beiersdorf, 1991); for sediment deposition environments the interplay between bottom currents and bathymetry plays an important role (Halbach, 1988). The depositional properties vary on a regional scale, considering large geomorphological terrain types, but are also impacted on a local scale of only a few kilometers and even less (Craig, 1979; Frazer and Fisk, 1981; Sharma and Kodagali, 1993; Mewes et al., 2014). Varying considerations of scale and regional differences in nodule exposure between different oceans across different studies have thus led to partly contradicting statements of the relationship between the Mn-nodule coverage/size and bathymetric settings.

Several investigations report small Mn-nodules and low coverages in depressions and plains which are considered as sediment accumulation sites, in contrast to seamounts, slopes and crests (Pattan and Kodagali, 1988; Sharma and Kodagali, 1993). Other studies discussed comparatively larger diagenetic Mn-nodules in plains which are also considered as sediment accumulation areas. More abundant but smaller hydrogenetic Mn-nodules have been observed in more rugged terrain (Skornyakova and Murdmaa, 1992; Kim et al., 2012; Widmann et al., 2014). Such terrains are interpreted to increase current velocities andd turbulences caused by channel effects reducing sediment accumulation.

(Mewes et al., 2014) present a correlation between Mn-nodule size and sedimentation rate, where large nodules correlate with a smaller amount of clay fraction in the sediments that they interpreted to be caused by stronger bottom currents/lower sedimentation rate. A similar observation is presented by (Skornyakova and Murdmaa, 1992) who state that diagenetic/large Mn-nodule formation is linked to a periodical redistribution of the surface sediment layer.

With respect to the large scale of the ship-based bathymetry in Figure 2, the working area of this study is located in a sediment-accumulating flat terrain with smooth bathymetry, characterized by the occurrence of medium to large (>4cm) Mn-nodules (Rühlemann et al., 2011). However, a more detailed view allows the identification of terrain variability on a scale of several tens to hundred meters that enables a more detailed assessment of the associated Mn-nodule coverage variability (Figure 5). Larger nodules/higher coverage values occur in depressions and at sloping seafloor when compared to broad scale bathymetry. Larger nodule sizes could be the result of stronger bottom currents preventing/reducing the deposition of sediment on nodules and/or favoring nodule growth. For another area in the German claim, box core (BC) samples taken by the Federal Institute of Geosciences and Natural Resources (BGR Hannover, Germany), revealed larger (diagenetic) nodules in a very broad scale flat terrain. This area has been classified by (Widmann et al., 2014) as an area of sediment accumulation and is compared to a rougher, supposedly sediment 'winnowing' area, with many smaller nodules formed hydrogenetically. The interpretation of sediment accumulating and winnowing areas is based on broad scale ship-based bathymetry of much coarser resolution compared to this study.

According to the study by (Skornyakova and Murdmaa, 1992) the data presented here indicate lower sedimentation rates associated with stronger bottom currents in the depressions supporting the growth of larger Mn-nodules. Increased bottom

currents within the depressions could possibly be induced by convergent channeling or turbulences of bottom currents, which contradicts the assumption of lower current strength and therefore higher sedimentation within depressions.

### 4.2.2 Local scale correlation between AUV-based bathymetry and Mn-nodule coverage/size

Variability in Mn-nodule coverage within several tens of meters or less can be correlated with AUV-based bathymetry. In sub-area A2, patches of low Mn-nodule coverage correlate with low bathymetric elevations even when the relief differs by less than one meter. The strongest correlation between low Mn-nodule coverage was determined with slightly convex shaped elevated structures (surfaces <1° slope, positive plan curvature and positive BPI values). These parameters most likely define local-scale sedimentation environment affecting the local balance between sediment accumulation and erosion The presented data show that favorable nodule growth/occurrence conditions coincide with gentle sloping sites and low relief depressions, where sediment is assumed to accumulate slowly.

Within sub-area A2 a smaller variability of Mn-nodule coverage can be observed in correlation with the 'Slope A2E' towards the East. This is in agreement with observations by (Sharma and Kodagali, 1993) who also observed more uniform nodule coverages in sloping areas. The authors point out that this could be a result of a larger exposure of the Mn-nodules rather than absolute difference, since they discovered discrepancies between direct sampling and results of photo analyses.

Rather special for the presented data set are the pronounced pit structures, observed throughout the AUV-mapped area with very little to no Mn-nodules observed at the sediment surface. This is in contradiction to the wider depressions, where a higher Mn-nodule coverage was observed. The existence of such pronounced depressions most likely leads to a reduction of bottom current velocities resulting in a higher sediment deposition of suspended sediment and potentially even sediment slumping from the sides. This could result in sedimentation rates too high for Mn-nodule formation (Halbach, 1988; Mewes et al., 2014) or a simple cover/burial of previously formed and still existing Mn-nodules below the sediment surface. The formation process of the pits is unclear, but could be karst structures (Kuhn et al., 2017), which are younger than the Mn-nodule formation which would point towards Mn-nodule burial within the pits. At the same time a higher sedimentation rate in a low current regime would also mean a higher accumulation of clay size particles, which are proposed to not be favorable for nodule growth (Mewes et al., 2014). Another possibility could be that these pit structures are pockmarks, formed by pore water release ((Harrington, 1985; Hovland and Judd, 1988) with a significant change in local pore water geochemical properties and eventually warmer temperature that prevented Mn-nodule formation in the past. Unfortunately the pit structures could not be sampled in more detail and it is unknown whether Mn-nodules exist at all or if different geochemical conditions are present within the pits. Similar, but larger structures exist in the DISCOL area (Greinert, 2015) showing very similar geochemical conditions as other Mn-covered areas in both highly detailed sediment surface analyses as well as deeper sediment cores.

### 4.2.3 Comparison between sub-areas A1 and A2

When comparing the relationships between the bathymetric derivatives and the Mn-nodule coverage it becomes evident, that correlations visible in A2 cannot be seen in A1 (Figure 6A, C), where areas of lower nodule coverage could not be matched with distinct terrain types. This result points towards additional parameters that influence Mn-nodule occurrence. Geochemical

processes could be involved that drive the Mn-nodule formation; these in turn depend on the sediment properties (composition, sedimentation rate, porosity, etc.). Bottom currents could additionally influence the sedimentation rate and affect the geochemical processes in the benthic boundary layer and Mn-nodule surface. Local differences in the hydrodynamic regime near the bottom seem likely, as the bathymetric derivatives vary between the two sub-areas. Sub-area A2 is bound towards East and North by elevated terrain (7m to 10m higher) which could have a focusing effect on bottom currents eventually causing a more erosional environment. In contrast, sub-area A1 is unbound by elevated terrain within 2km distance. This might cause a stronger influence of seasonally changing bottom currents, preventing a clearer correlation of Mn-nodule coverage with the seafloor morphology.

#### 4.2.4 Broad- vs. small-scale correlation

The observations made on broad scale (several hundreds of meters; grid cell size of 55m) show that high Mn-nodule coverage correlates with depressions (Figure 5) which is consistent with observations on smaller-scale (scale of tens of meters, grid cell size of 5.5m) for sub-area A2 (Figure 7). Outside of A2 decreasing Mn-nodule coverage correlates with steeper sloping areas, which is contradicting to observations on small-scale, where the lowest Mn-nodule coverages correlate with extremely low slope angles of less than $1.8°$. This contradicting finding highlights that simple and generalized correlations between Mn-nodule occurrence and bathymetric but also geochemical properties in the sediment might not be possible on regional scale (10km to 1000km) but on local scale (100m - 10km). This is because the formation parameters also change on such local scales which are not possible to accurately predict using ship-based multibeam data, 'sparse' box-coring (distances of few kilometers) and limited information about current regimes.

### 4.3 Sediment plume re-settling

The approach of conducting a photo mosaic survey before and after a seafloor disturbance proved successful for detecting sediment blanketing visually, offering the possibility to accurately map the area of strongest plume-impact. This area is characterized by the sediment plume transport direction and re-settling of the majority of the sediment. Very fine particles within the sediment plume might be dispersed much further; more detailed biological studies need to evaluate which sediment concentrations and grain sizes will impact benthic organisms on long time scales (cumulative effects) outside the visually clearly detectable impacted areas.

The thickness of the resettled sediments could not be determined from the AUV-based images or ROV-based video footage during the cruises. Video observations from other, similar areas point towards a sediment cover on millimeter or sub-millimeter range that can still be detected in images (e.g. in laboratory experiments (Yamazaki et al., 1997)).

#### 4.3.1 Morphology-influence on sediment transport

The extent of the visible sediment blanketing, that varies over several tens of meters, can be related to a focusing of the sediment plume settling or the prevention of it through small-scaled morphological changes in form of barriers (steeper slopes

facing against the current) or the opening of plume transportation pathways (sloping terrain with the current). Varying terrain in general will modify the current regime near the bottom and thus the settling properties of the sediment plume; it might also enhance the interactions between the particles due to increased turbulences that might stimulate increased flocculation and

5 thus scavenging of very small particles that otherwise would be much further distributed. The shorter transport of sediment in north- and southward direction from the EBS track along the 'Slope A2E' implies that the transportation of the suspension load follows the slope downhill. In 'Sub-section A2W', where the terrain is very smooth (the relief changes by 1m to 2m) a dependency of the sediment blanketing extent to structures of the undulating seafloor could still be observed. At the western end of Sub-section A2W the east-facing slopes act as barrier for an undisturbed migration of the sediment plume with the bottom

current towards the South. The spreading of the sediment blanketing is wider in the East of Sub-section A2W where the seafloor is almost horizontal, before slightly dipping towards the East and into the Pit 1 structure. The slopes considered show angles of less than 2° and the morphological variability is sometimes less than 1m. More distinct features, like Pit 1 (Figure 9), cause a more variable sediment plume dispersal. The sediment blanketing within this 2m deep feature does not exceed the southward edge of the depression. The re-suspended sediment seems trapped within this feature with possible additional suspension load

coming from the neighboring eastward slope.

### 4.3.2 Estimation of plume height

In a first approach we estimated the plume height generated by the EBS by considering the extent of the observed sediment blanketing and measured bottom current velocities at the time of the EBS deployment (31mm/s; measured by ADCP). Former models from the CCZ reported settling velocities of particles in a sediment plume in the range of 0.1 to 1mm/s derived from

20 visual and experimental data (Lavelle et al., 1981; Oebius et al., 2001). Preliminary results of particle size analysis from a comparable site within the PA 1 indicate a median grain size of 29$\mu$m (Benjamin Gillard, Jacobs University Bremen, Germany, personal comm.). Following Stokes' law and disregarding aggregation of the particles, the determined median particle size for the area would translate to sinking velocities of approximately 1mm/s. Assuming an average dispersal width of 30-50m downstream, as indicated for the A2W sub-section (Figure 9), this would require a plume height of approximately 0.96 to 1.6m.

Aggregation processes leading to larger particle sizes are likely to occur which, due to increased friction, would sink slower than similar sized 'Stokes' particles but that would scavenge a substantial amount of very small particles (personal comm. Laurenz Thomsen, Jacobs University Bremen, Germany). As part of studies in the south Pacific DISCOL area, lander-based ADCP backscatter measurements detected a passing-by sediment plume induced by a similar EBS experiment as discussed here. These data indicate a plume height between 1.5m to 2m (Greinert, 2015).

### 4.3.3 Implications for possible mining scenario

It can be assumed that, due to the higher turbulences caused by the deployment of an industrial collector system and the continuous release of suspended material into the water column during mining, the dynamic behavior of the sediment plume could be altered and adjusted in such a way that the suspended sediment is re-settling in the fastest possible way, keeping the dispersion to a minimum. Determining the dynamic behavior of the plume under different collector-dispersion scenarios by

monitoring in-situ and under real-mining conditions is thus essential to improve our understanding and model capacity with regards to the near- and far-field plume distribution and finally to evaluate ecological short- and long-term impacts.

These ecological impacts can be significantly confined to a small area by reducing the height of the sediment plume, increasing the settling velocity and aggregation of particles (scavenging the very fine sediment fraction. Vertical discharge of sediment after its separation from the Mn-nodules should be avoided; instead a horizontal discharge close to the bottom (<10m from the bottom; below 'stable' stratification above the well mixed bottom boundary layer) with a velocity as slow as possible (speed of the collector) should be aimed for. One first implementation of this concept was the setup of the DSSRS disturber (Brockett and Richards, 1994) deployed in a few large-scale Benthic Impact Experiments (BIE-II, JET, IOM-BIE, INDEX).

As indicated by our results, a low-height sediment plume will be trapped in small depressions. Thus a detailed knowledge of the local morphology on small scales is a pre-requisite to correctly determine the area and thickness of re-settling sediment. This is also relevant in planning adjacent mining tracks from a miner's point of view, since strong sediment blanketing might burry adjacent nodules to be mined. According to our results, this impact will be highest in sediment accumulation sites, but even on 'flat' areas with slopes of less than $3°$ the distribution of the sediment plume and the resulting sediment blanketing distance will vary on a range of several tens of meters. In areas with steeper slopes (e.g. $10°$), the sediment blanketing distance can be even wider.

In our very small scale experiment, the EBS created a local impact with clearly visible sediment blanketing within 100m downstream off the track. This localized impact is also the result of only partial re-suspension of the surface sediments that was directly caused by the EBS (1.2 in width). Observations of EBS tracks during another experiment revealed that a larger part of the sediment is compressed by the EBS and pushed aside with only a smaller (unknown) fraction being suspended (Boetius, 2015). It can be speculated that re-sedimentation of particles outside the visible blanketing area is minor, will happen over longer time, and thus might not have a significant effect on the benthic organisms and the ecosystem (short and long-term cumulative impacts on specific fauna still needs to be determined).

The actual scenario of disturbance will be different during real-case mining during which the top 10 - 20cm of the sediment are removed, then 'filtered for nodules' and are then discharged at the seafloor. One single track will be about 17m wide as e.g. planned in a German concept (Kuhn et al., 2011), whereas the track width of the EBS was only 1.2m. As not only one track will be mined, but the collector system will operate constantly in 'lawn-mowing' pattern of long tracks scraping off the seafloor surface, the entire mined area will see a strong impact (Jankowski and Zielke, 2001). Considering local topography, bottom currents, optimizing particle settling (You, 2004) for fast and effective flocculation by the collector, and the cleaning of the sediment plume from the water column by settling phytodetritus from plankton blooms (increased flocculation) the size of the impacted area and the impact itself caused by the sediment blanketing outside the mined area might be rather small (<10km) and controllable. For a final validation, an experimental setup closer to the expected mining conditions is needed (Sharma, 2011); the presented study shows that we have the understanding, tools and the methodologies at hand to perform monitoring studies needed for such a realistic deep sea mining experiment.

# 5 Conclusions

We conclude that for both of our study topics, the Mn-nodule distribution to terrain comparison as well as the re-deposition of sediments indicate that Mn-nodule coverage and sediment blanketing vary measurably on very small scale (several tens to hundreds of meters), even if the seafloor terrain changes are minor (less than 1m vertical change). This supports the second conclusion: that spatial scale needs to be considered when discussing possible parameters that influence Mn-nodule coverage as such, and that relations found in one region most likely cannot be generalized to other regions and across different scales. Confirming previous studies, our data also show no simple relationship between Mn-nodule coverage with the seafloor morphology even when working on the same spatial scale. It needs to be realized that a complex interrelation between morphological characteristics and local environmental conditions (physical, chemical, sedimentological) influence the visually detectable Mn-nodule coverage at the seafloor surface. Reasons for this are that 1) variable amounts of Mn-nodules have formed under different geochemical, bottom current or sedimentological conditions in different places, 2) Mn-nodules might have dissolved in certain areas because of changing geochemical conditions or mechanical erosion, 3) Mn-nodules were buried by sediment whereas the sediment deposition pattern is influenced by the seafloor terrain and its interplay with bottom currents and 4) the existence of Mn-nodules (abundance, size, total coverage) itself influences sediment erosion, making denser covered areas with large nodules more resistant against sediment erosion.

With respect to the sediment plume study it became obvious that a visible blanketing occurs in a limited distance (here <100m) away from the disturbance track and that the blanketing pattern strongly depends on bottom current direction, strength, small-scale bathymetry and initial plume height. From these observations it can be concluded that each sediment plume disposal via an exhaust/diffusor of the collector should occur horizontally as close to the bottom as possible, rather than on top of the vehicle, blowing the sediment particles high into the water column, aiming for a finer dispersal over larger areas. Our studies also highlight that the performed disturbance experiment cannot be scaled up to a real mining scenario and that more detailed studies are required to understand and quantify the cumulative impact of unsettled particles on filter-feeding organisms beyond the clearly visual blanketing area.

On the technical side the study showed that we have the needed tools and techniques at hand to map the seafloor for Mn-nodule resource assessments and a better understanding of Mn-nodule distribution as well as for assessing mining impacts visually. It became clear that without such high resolution techniques valid assessments cannot be carried out. Areas that appeared suitable of mining (slopes $\leq 3°$ in ship-based bathymetric data showed steeper relief (slopes $>3°$) in higher resolution AUV-based data. For an 'environmentally friendly deep sea mining' such high resolving maps are a pre-requisite to accurately define areas that need to be protected, and maneuver mining infrastructure around them considering the actual bottom currents and sediment settling areas during the mining.

*Code and data availability.* Source code for the automated nodule delineation is available in Pangaea (Schoening, 2017c). The data used in this work is available in Pangaea. This includes MBES data (Greinert, 2016), optical imagery (Greinert et al., 2017) and image-derived nodule coverages (Schoening, 2017a, b).

## Appendix A: Methodology

### A1    Calculation of the bathymetric derivatives

Slope was calculated using the algorithm included in the 'Spatial Analyst' Toolbox (Burrough, 1986) of ArcGIS (Table A4
output in 'degrees'). Curvature as second order derivative of the bathymetry represents the slope of the slope. It has also been
determined with the ArcGIS 'Spatial Analyst' Toolbox. For each cell a 4th order polynomial is fit to a surface composed
of a 3x3 cell window. From this surface the tool calculates the coefficients (Table A3), which are set into relation with the
elevation values for every cell (Zevenbergen and Thorne, 1987). The two maximum slope dependent curvature values of the
plan curvature, perpendicular to the maximum slope, and the profile curvature parallel to the maximum slope direction were
calculated. The plan curvature defines flow convergence (concave surface, values < 0) and divergence (convex values > 0).
Profile curvature affects the acceleration of the flow with values < 0 indicating a concave shaped surface and values > 0
indicating convex shaped surface. In addition the total curvature of a surface has been calculated which is also > 0 when
convex shaped or < 0 when concave shaped. Curvature values of 0 are indicative for flat surfaces.

For calculating aspect, Bathymetric Position Index (BPI) and Terrain Ruggedness the ArcGIS 'Benthic Terrain Modeler'
(BTM) Add-in (Wright et al., 2012) and its incorporated algorithms were used. The classification performed by the BTM
is based on manually set properties (Tables A1 & A2) of the derivatives slope, BPI (fine scale and broad scale) and water
depth. This simple classification process provides sufficient information to distinguish different terrain settings of the study
area (Figures 2 & 3). The classification of the AUV-mapped study area (Figure 6, Table A5) was performed to reveal areas of
lower Mn-nodule coverage and is based on the derivatives BPI, slope and plan curvature, since these morphological parameters
showed the best correlation with Mn-nodule coverage (Figure 7). For the BPI calculation of the AUV-mapped area the algorithm
used within the BTM was modified (without integer rounding) to preserve the small-scaled features (Wilson et al., 2007).

The aspect is defined as the inclination direction of the maximum rate of change in depth from each cell to its neighbors,
the slope inclination (Burrough, 1986). The algorithm calculates an aspect value for each cell of a raster and incorporates the
respective adjacent cells in both horizontal directions from the center cell (dz/dx and dz/dy) (Table A4).
The BPI describes the relative topographic variability of a central grid cell to a circular annulus with an inner and outer
radius, both are manually defined (Table A1) (Weiss, 2001; Wright et al., 2012). For classification a broad BPI (BBPI; large
radii) and a fine-scale BPI (FBPI, small radii) are calculated and standardized. Positive values indicate that the central grid cell
is elevated with respect to the mean annulus height values, negative values indicate depressions. The BPI is usually subscribed
with the applied scale factor (grid cell size × outer annulus radius).
The Terrain Ruggedness was calculated for the AUV bathymetric data set (Figure A4) using the algorithm for the Vector
Terrain Measurement (VRM) of the BTM (Sappington et al., 2007). It incorporates slope and aspect heterogeneity of the terrain
and is defined as the magnitude of a resultant normalized vector from the decomposed x, y, z components of the cells and their
slope and aspect, normalized to the number of cells in the neighborhood (Sappington et al., 2007). The Terrain Ruggedness is a
unit less measure, ranging from 0 (flat) to 1 (most rugged). All derivative results were displayed and evaluated in ArcGIS 10.2.

| Area | | Bathymetry | Slope | Aspect | B-BPI | F-BPI | VRM |
|---|---|---|---|---|---|---|---|
| PA I (Fig. 2) | Grid cell size | 110m | 110m | | 110m | 110m | |
| | inner radius | | | | 10 | 10 | |
| | outer radius | | | | 100 | 100 | |
| | scale factor | | | | 11000 | 55000 | |
| Working Area (Fig. 3) | Grid cell size | 55m | 55m | 55m | 55m | 55m | 55m |
| | inner radius | | | | 20 | 10 | |
| | outer radius | | | | 200 | 50 | |
| | scale factor | | | | 11000 | 2750 | |
| AUV MB (Fig. 4) | Grid cell size | 5m | 5.5m | 5.5m | 5.5m | 5.5m | 5.5m |
| | inner radius | | | | 2 | | |
| | outer radius | | | | 80 | | |
| | scale factor | | | | 440 | | |
| | Neighborhood (cells) | | | | | | 3x3 |

**Table A1.** Metadata of the created maps including raster cell sizes for the considered regions and sub-regions.

## A2 Interpretation of Mn-nodule size results

Considering a potential error in correctly detecting nodules by the CoMoNoD algorithm, the application of quantiles of the size distribution allows a more robust interpretation of the data. It is suggested not to use size values of smallest and largest 1% of the quantile calculation, due to the above mentioned error source. The graph in Figure A1 illustrates the quantiles of the calculated sizes of two images, which clearly differ from each other. The graph correctly displays a size difference between both images, indicating more larger nodules for image 29302. This shows that applying CoMoNoD to calculate nodule sizes is reasonable. In this case, the best differentiation exists for the 50% - 75% quantile. Towards larger and smaller size values the two curves approach each other which points towards the detection of similar - none nodule - features in both images. Therefore, the median size values are considered to best represent the Mn-nodule size distribution differences between images/areas. Without ground-truth data from sampling, computed size values should not be used as absolute values for resource assessment. However they can be used to quantify nodule size distributions within seafloor areas and hence to compare variations in nodule distribution and coverage.

*Competing interests.* The authors declare no competing interests.

| Class | Zone | BroadBPI Lower | BroadBPI Upper | FineBPI Lower | FineBPI Upper | Slope Lower | SLope Upper | Depth Lower | Depth Upper |
|---|---|---|---|---|---|---|---|---|---|
| 1 | seamounts | 80 | | | | | | | |
| 2 | mineable ridges | 40 | 80 | | 100 | 0° | 3° | -4110m | |
| 3 | ridges | 40 | 80 | | 100 | | | -4110m | |
| 4 | mineable plateau | -20 | 40 | | | 0° | 3° | -4200m | -4110m |
| 5 | elevated plateau | -20 | 40 | | | | | -4200m | -4110m |
| 6 | flat depression | -70 | -10 | | | 0° | 3° | -4300m | -4200m |
| 7 | depression | -70 | -10 | | | | | -4280m | -4200m |
| 8 | mineable deep depression | -160 | -70 | -230 | -10 | 0° | 3° | -4500m | -4280m |
| 9 | deep depression | -160 | -70 | -230 | -10 | | | -4500m | -4280m |

**Table A2.** Classification dictionary with upper and lower bounds for the classification of the PA I area used with the BTM.

| Class | Zone | BroadBPI Lower | BroadBPI Upper | FineBPI Lower | FineBPI Upper | Slope Lower | SLope Upper | Depth Lower | Depth Upper |
|---|---|---|---|---|---|---|---|---|---|
| 1 | mineable elevations | 70 | | | | 0° | 3° | -4110m | -4050m |
| 2 | un-mineable elevations | 70 | | | | | | -4110m | -4050m |
| 3 | mineable minor elevations | -20 | 70 | | | 0° | 3° | -4130m | -4110m |
| 4 | un-mineable minor elevations | -20 | 70 | | | | | -4130m | -4110m |
| 5 | mineable small scale basins | | | -130 | -40 | 0° | 3° | -4150m | -4130m |
| 6 | un-mineable small scale basins | | | -130 | -40 | | | -4150m | -4130m |
| 7 | mineable depressions | -80 | -20 | | | 0° | 3° | -4140m | -4125m |
| 8 | un-mineable depressions | -80 | -20 | | | | | -4140m | -4125m |
| 9 | mineable deep depressions | -280 | -80 | | | 0° | 3° | | -4140m |
| 10 | un-mineable deep depressions | -280 | -80 | | | | | | -4140m |

**Table A3.** Classification dictionary with upper and lower bounds for the classification of the working area used with the BTM.

| Bathymetric derivative | ArcGIS Tool | Algorithm | Literature |
|---|---|---|---|
| Slope | Spatial Analyst | slope_degree = $\text{atan}(\sqrt{(dz/dx)^2 + (dz/dy)^2}) \times 360°/2\pi$ | (Burrough, 1986) |
| Aspect | BTM | aspect = $\text{atan2}(dz/dy, dz/dx) \times 360°/2\pi$ | (Burrough, 1986) |
| BPI | BTM | BPI[scalefactor] = int((bathy-focalmean(bathy,annulus,irad,orad))+0.5) | (Weiss, 2001) |
| BPI (AUV) | Raster Calculator | BPI[scalefactor] = 'grid'-focalmean('grid',circle,r) | (Wilson et al., 2007) |
| BPI_Std | BTM | BPI[scalefactor]_std = int(((BPI<scalefactor>-mean/stddev) $\times$ 100)+0.5 | (Weiss, 2001) |
| VRM | BTM | VRM = $1 - \sqrt{(\sum x)^2 + (\sum y)^2 + (\sum z)^2}/n$ | (Sappington et al., 2007) |
| Curvature | Spatial Analyst | K = $(\delta^2 Z/\delta S^2)/(1+\delta Z/\delta S)^2)^{3/2}$ | (Zevenbergen and Thorne, 1987) |
| Plan Curvature | Spatial Analyst | $2 \times (D\sin^2\theta + E\cos^2\theta - F\sin\theta\cos\theta)$ | (Zevenbergen and Thorne, 1987) |
| Profile Curvature | Spatial Analyst | $-2 \times (D\cos^2\theta + E\sin^2\theta - F\sin\theta\cos\theta)$ | (Zevenbergen and Thorne, 1987) |

**Table A4.** Algorithms and ArcGIS Tools applied for the calculation of the bathymetric derivatives.

| Class | Zone | FineBPI50st Lower | FineBPI50st Upper | Slope Lower [deg] | Slope Upper [deg] | Plan Curvature Lower [radians/m] | Plan Curvature Upper [radians/m] |
|---|---|---|---|---|---|---|---|
| 1 | lower coverage | 0 | | 0 | 1.8 | -0.02 | |
| 2 | higher coverage | -200 | 0 | 1.5 | | | -0.02 |
| 3 | no nodules | | -200 | 0 | 2 | | -0.1 |

**Table A5.** Classification dictionary for the classification of the AUV-mapped study area used with the BTM, to reveal areas of possible lower Mn-nodule coverage.

| | Aspect [deg] | | Slope [deg] | | BPI440st | | VRM | |
|---|---|---|---|---|---|---|---|---|
| | A1 | A2 | A1 | A2 | A1 | A2 | A1 | A2 |
| Mean | 163 | 220 | 1.28 | 1.48 | -24 | -54 | 9.98E-06 | 8.12E-06 |
| Median | 177 | 251 | 0.97 | 1.28 | -13 | -61 | 3.09E-06 | 3.93E-06 |
| Mode | 180 | 225 | 0.76 | 1.50 | -6 | -97 | 1.25E-07 | 1.25E-06 |
| Standard Deviation | 64 | 87 | 0.93 | 0.93 | 68.67 | 58.98 | 2.44E-05 | 1.79E-06 |
| Range | 360 | 361 | 4.92 | 4.34 | 360 | 316 | 3.31E-04 | 3.23E-04 |
| Minimum | 0 | 0 | 0.03 | 0.03 | -250 | -225 | 0 | 0 |
| Maximum | 360 | 360 | 4.95 | 4.37 | 110 | 91 | 3.31E-04 | 3.23E-04 |
| Count | 10120 | 16890 | 10120 | 16890 | 10120 | 16890 | 10120 | 16890 |

**Table A6.** Statistics summary of the derivatives derived from the AUV-obtained bathymetry for the A1 and A2 survey areas.

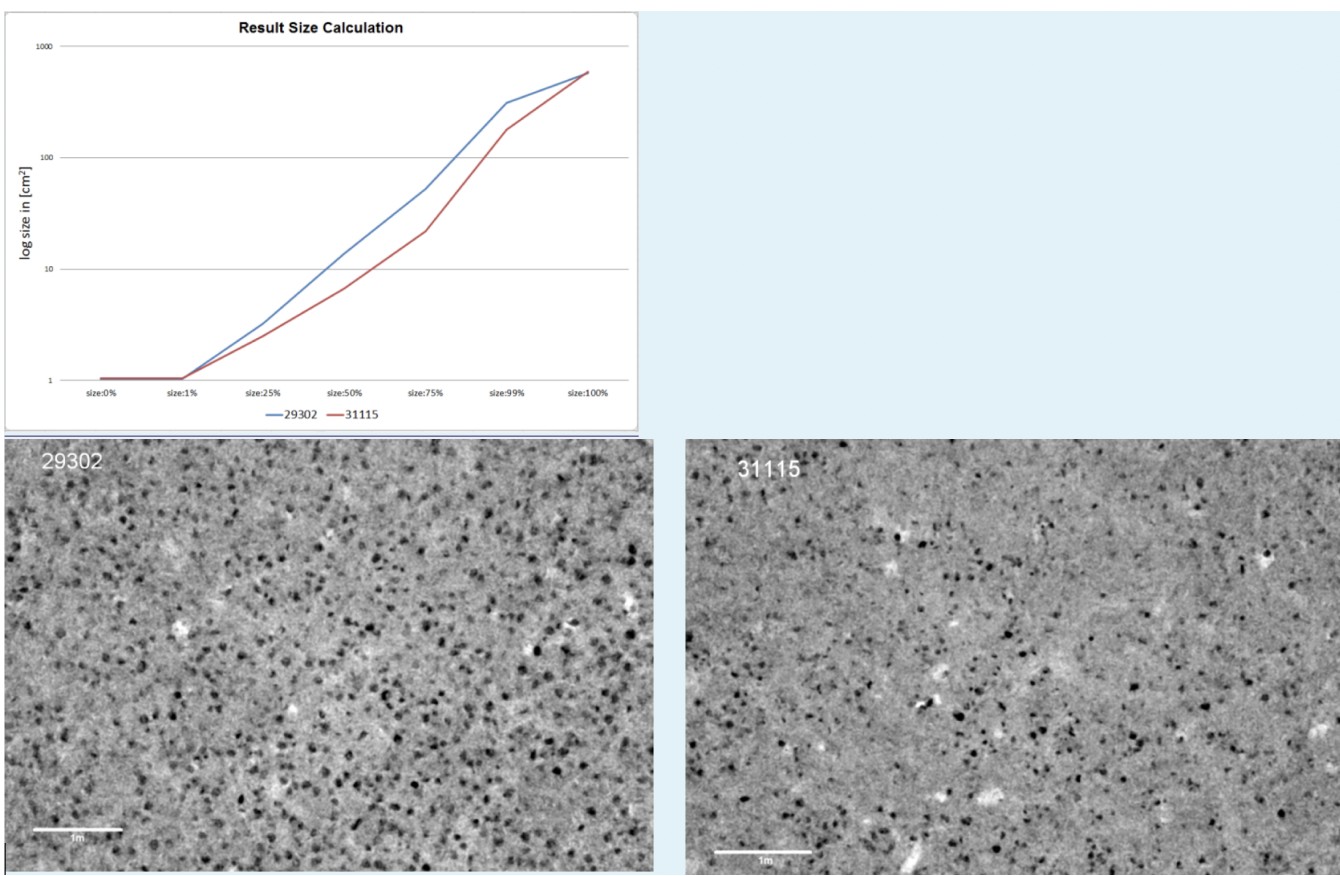

**Figure A1.** Two example images (bottom) which clearly differ in nodule size and coverage. The graph shows the size distribution as calculated by the CoMoNoD algorithm. The most significant difference is observed in the 75% quantile.

*Acknowledgements.* We thank the Captain and crew of RV SONNE SO239 for their cooperation and valuable contribution to a successful cruise. All data were acquired within the framework of the JPIO Project 'Ecological Aspects of Deep-Sea mining (D1753)'; financial support was also provided by the EU project MIDAS (FP7, Grant Agreement No. 603418). The Federal Institute for Geosciences and Natural Resources - BGR is to be acknowledged for sharing data and for their valuable input regarding industrial and resource assessment development. Benjamin Gillard (PhD at Jacobs University Bremen, Germany) is to be thanked for providing preliminary results of his studies regarding particle size distributions and settling behavior within the area studied here. We express our gratitude to the GEOMAR AUV team for their splendid support and professional attitude during the cruise. This is publication 33 of the DeepSea Monitoring Group at GEOMAR;

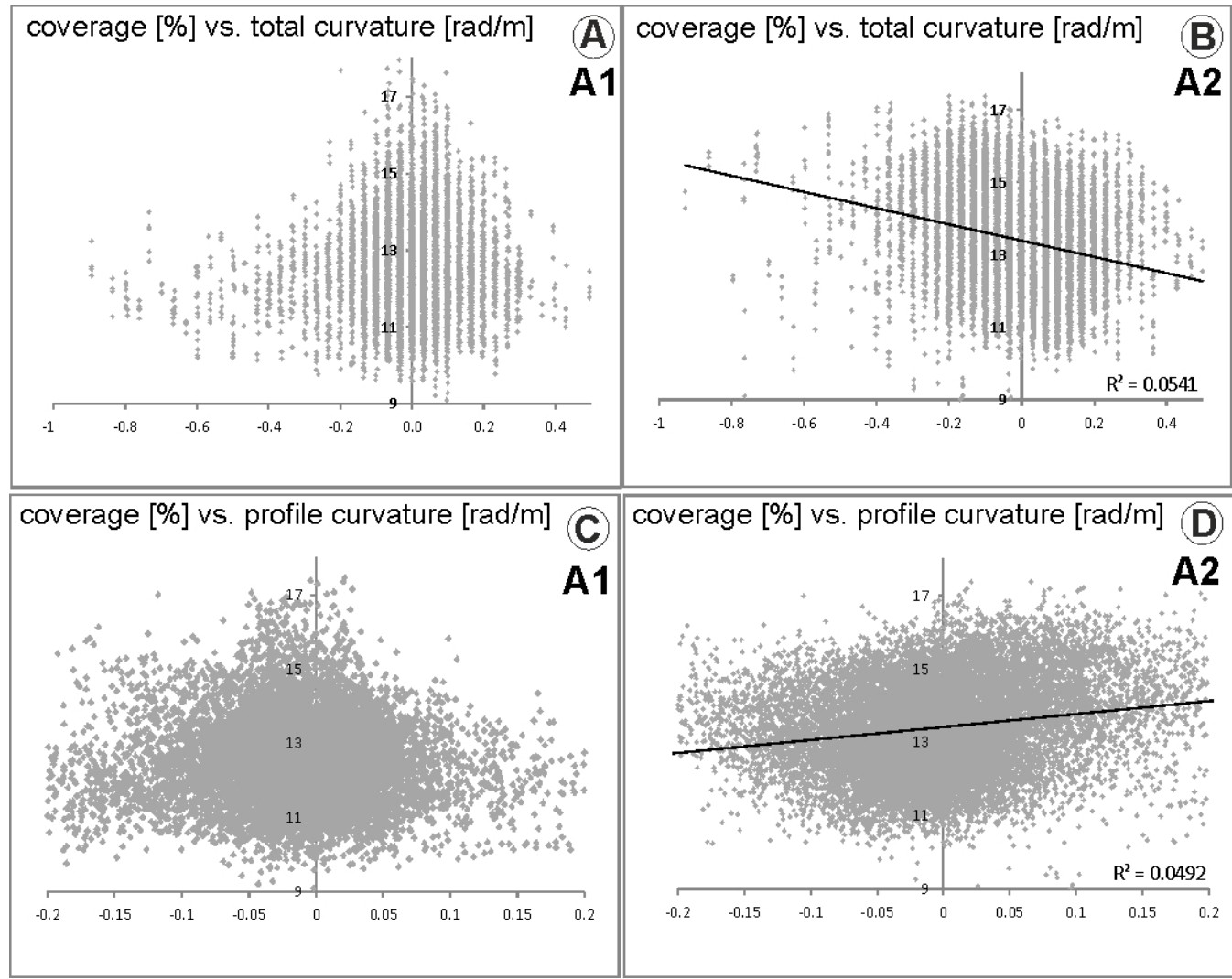

**Figure A2.** Scatterplots indicating the relation between Mn-nodule coverage and total curvature (A,B) and profile curvature (C,D) in the A1 and A2 sub-areas. Only in area A2 weak correlations could be observed.

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

Boetius, A.: RV SONNE Fahrtbericht/Cruise Report SO242-2 [SO242/2]: JPI OCEANS Ecological Aspects of Deep-Sea Mining, DISCOL Revisited, Guayaquil-Guayaquil (Equador), 28.08.-01.10. 2015, 2015.

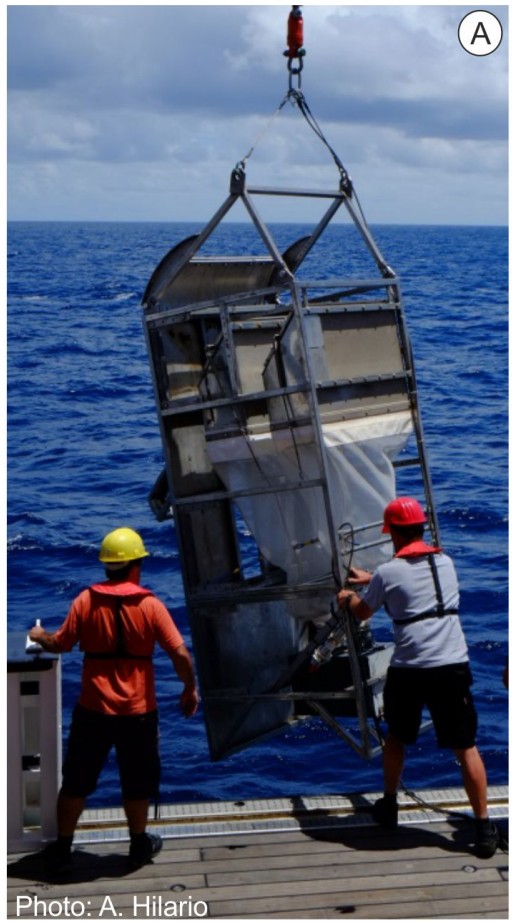

Photo: A. Hilario

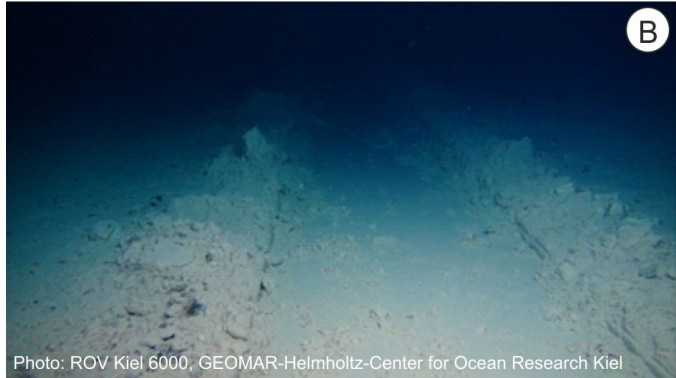

Photo: ROV Kiel 6000, GEOMAR-Helmholtz-Center for Ocean Research Kiel

**Figure A3.** A) The Epibenthic sledge that created the monitored sediment plume (Photo: A. Hilario) (Brenke, 2005). B) Track created by the EBS at the seafloor, approximately 20 cm deep and 1.5 m wide Photo: ROV Kiel 6000, GEOMAR Helmholtz-Center for Ocean Research Kiel.

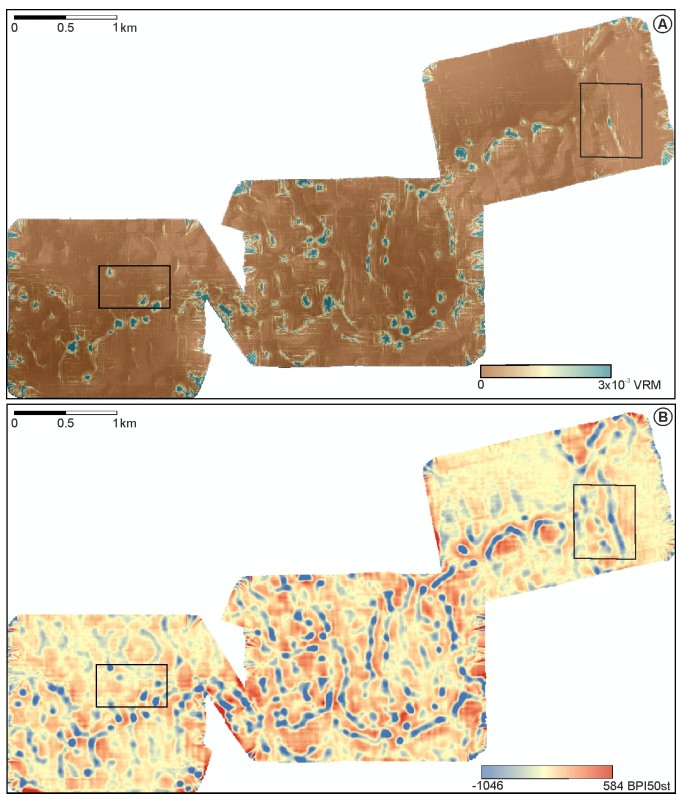

**Figure A4.** A) Terrain Roughness-indicating map derived from AUV-acquired bathymetry data (calculated with the VRM algorithm (Sappington et al., 2007). B) BPI50st derived from the AUV-obtained bathymetric data set; the lowest values indicate areas with 'pit-structures' that are observed throughout the entire area.

Brenke, N.: An epibenthic sledge for operations on marine soft bottom and bedrock, Marine Technology Society Journal, 39, 10–21, 2005.

Brockett, T. and Richards, C. Z.: Deepsea mining simulator for environmental impact studies, Sea Technology, 35, 77–82, 1994.

Burns, R.: Observations and measurements during the monitoring of deep ocean manganese nodule mining tests in the North Pacific, March-

5    May 1978, vol. 47, US Department of Commerce, National Oceanic and Atmospheric Administration, Environmental Research Laboratories, 1980.

Burrough, P. A.: Principles of geographical information systems for land resources assessment, 1986.

Craig, J. D.: The relationship between bathymetry and ferromanganese deposits in the north equatorial Pacific, Marine Geology, 29, 165–186, 1979.

10   Edwards, B. D., Dartnell, P., and Chezar, H.: Characterizing benthic substrates of Santa Monica Bay with seafloor photography and multibeam sonar imagery, Marine Environmental Research, 56, 47–66, 2003.

Foell, E., Thiel, H., and Schriever, G.: A LONG-TERM, LARGE-SCALE, DISTURBANCE-RECOLONISATION EXPERIMENT IN THE ABYSSAL EASTERN TROPICAL SOUTH PACIFIC OCEAN, 1990.

Frazer, J. and Fisk, M. B.: Geological factors related to characteristics of sea-floor manganese nodule deposits, Deep Sea Research Part A. Oceanographic Research Papers, 28, 1533–1551, 1981.

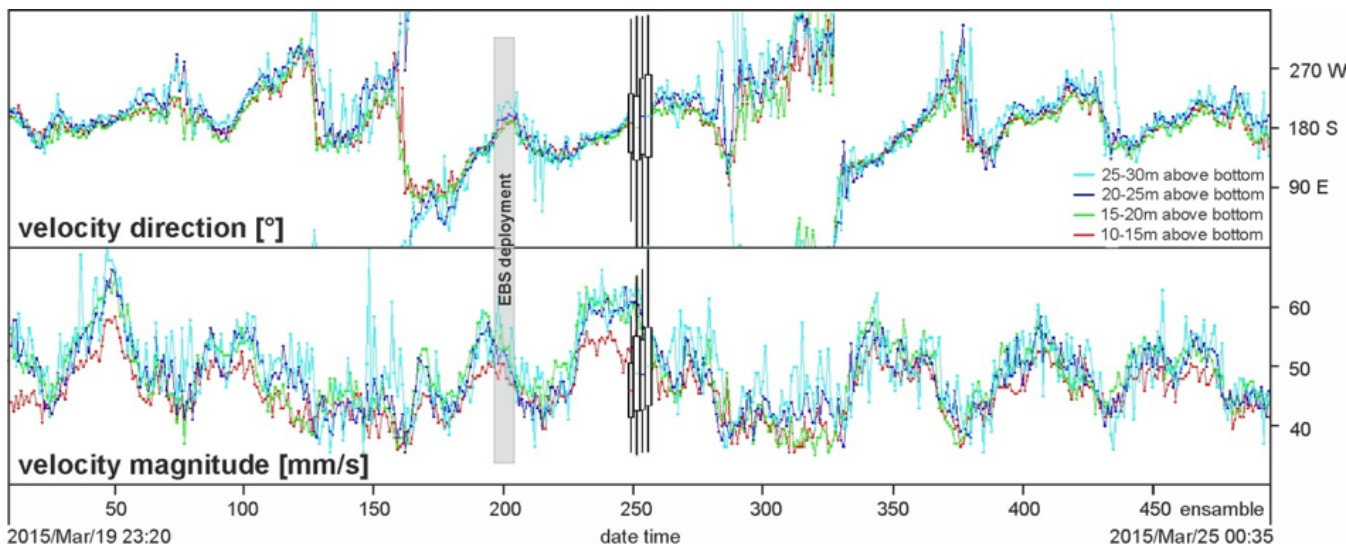

**Figure A5.** ADCP data obtained by while a DOS lander deployment during SO239 indicating a SSW current flow while the EBS deployment (grey shaded box; ensemble 196 to 204). Box plots show the mean and standard deviation of the time series of the entire deployment. Mean currents between 10 and 30 m above the bottom are towards the south with about 50 mm/s.

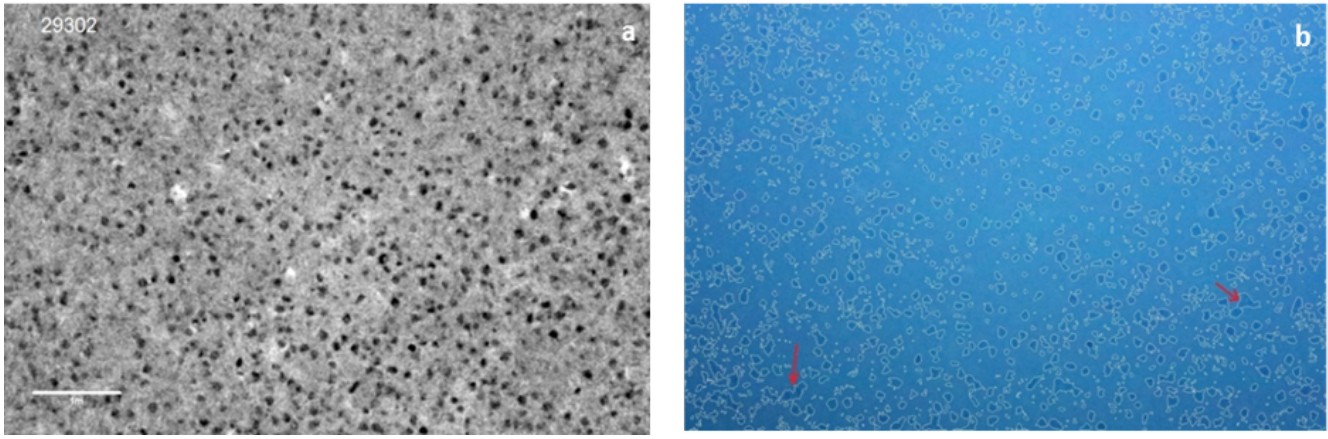

**Figure A6.** Example for how the program identifies nodules in the image. a) The greyscale image provides higher contrasts for better nodule identification. b) Image showing boundaries of the nodules the program has set for each recognized nodule. Red arrows indicate examples for locations where several nodules are bound together, which leads to larger values in the size calculation. The white scale bar in sub-figure a corresponds to 1m.

Fukushima, T. et al.: Overview" Japan Deep-Sea Impact Experiment= JET", in: First ISOPE Ocean Mining Symposium, International Society of Offshore and Polar Engineers, 1995.

Greinert, J.: RV SONNE Fahrtbericht/Cruise Report SO242-1 [SO242/1]: JPI OCEANS Ecological Aspects of Deep-Sea Mining, DISCOL Revisited, Guayaquil-Guayaquil (Equador), 28.07.-25.08. 2015, GEOMAR Reports, 2015.

Greinert, J.: Swath sonar multibeam EM122 bathymetry during SONNE cruise SO239 with links to raw data files., https://doi.org/10.1594/PANGAEA.859456, https://doi.pangaea/10.1594/PANGAEA.859456, 2016.

Greinert, J., Schoening, T., Köser, K., and Rothenbeck, M.: Seafloor images and raw context data along AUV tracks during SONNE cruises SO239 and SO242/1, https://doi.pangaea.de/10.1594/PANGAEA.882349, 2017.

Halbach, P.: The manganese nodule belt of the Pacific Ocean: Geological environment, nodule formation and mining aspects, Enke, 1988.

Harrington, P.: Formation of pockmarks by pore-water escape, Geo-Marine Letters, 5, 193–197, 1985.

Hovland, M. and Judd, A.: Seabed pockmarks and seepages: impact on geology, biology, and the marine environment, Springer, 1988.

Jankowski, J., Malcherek, A., and Zielke, W.: Numerical modeling of suspended sediment due to deep-sea mining, Journal of Geophysical Research: Oceans, 101, 3545–3560, 1996.

Jankowski, J. A. and Zielke, W.: The mesoscale sediment transport due to technical activities in the deep sea, Deep Sea Research Part II: Topical Studies in Oceanography, 48, 3487–3521, 2001.

Jeong, K., Kang, J., and Chough, S.: Sedimentary processes and manganese nodule formation in the Korea Deep Ocean Study (KODOS) area, western part of Clarion-Clipperton fracture zones, northeast equatorial Pacific, Marine Geology, 122, 125–150, 1994.

Jones, A. et al.: Review of benthic impact experiments related to seabed mining, in: Offshore Technology Conference, Offshore Technology Conference, 2000.

Jung, H.-S., Ko, Y.-T., Chi, S.-B., and Moon, J.-W.: Characteristics of seafloor morphology and ferromanganese nodule occurrence in the Korea deep-sea environmental study (KODES) area, NE Equatorial Pacific, Marine georesources & geotechnology, 19, 167–180, 2001.

Kim, J., Hyeong, K., Lee, H.-B., and Ko, Y.-T.: Relationship between polymetallic nodule genesis and sediment distribution in the KODOS (Korea Deep Ocean Study) area, northeastern Pacific, Ocean Science Journal, 47, 197–207, 2012.

Kotlinski, R., Stoyanova, V., et al.: Physical, Chemical, And Geological Changes of Marine Environment Caused By the Benthic Impact Experiment At the 10M BIE Site, in: The Eighth International Offshore and Polar Engineering Conference, International Society of Offshore and Polar Engineers, 1998.

Kuhn, T.: Cruise Report SO 240 - FLUM: Low-temperature fluid circulation at seamounts and hydrothermal pits: heat flow regime, impact on biogeochemical processes, and its potential influence on the occurrence and composition of manganese nodules in the equatorial eastern Pacific, https://doi.org/10.2312/cr_so240, 2015.

Kuhn, T. and Rathke, M.: 2017.

Kuhn, T., Rühlemann, C., Wiedicke-Hombach, M., Rutkowsky, J., Wirth, H., Koenig, D., Kleinen, T., and Mathy, T.: Tiefseeförderung von Manganknollen, Schiff & Hafen, 5, 78–83, 2011.

Kuhn, T., Versteegh, G., Villinger, H., Dohrmann, I., Heller, C., Koschinsky, A., Kaul, N., Ritter, S., Wegorzewski, A., and Kasten, S.: Widespread seawater circulation in 18–22 Ma oceanic crust: Impact on heat flow and sediment geochemistry, Geology, 45, 799–802, 2017.

Kwasnitschka, T., Köser, K., Sticklus, J., Rothenbeck, M., Weiß, T., Wenzlaff, E., Schoening, T., Triebe, L., Steinführer, A., Devey, C., et al.: DeepSurveyCam - a deep ocean optical mapping system, Sensors, 16, 164, 2016.

Lavelle, J., Ozturgut, E., Swift, S., and Erickson, B.: Dispersal and resedimentation of the benthic plume from deep-sea mining operations: a model with calibration, Marine Mining, 3, 59–93, 1981.

Linke, P. and Lackschewitz, K.: Autonomous Underwater Vehicle ABYSS, Journal of large-scale research facilities JLSRF, 2, 79, 2016.

Markussen, J. M.: Deep Seabed Mining and the Environment: Consequences, Perceptions, and Regulations, Green Globe Yearbook of International Co-operation on Environment and Development, pp. 31–39, 1994.

Martínez Arbizu, P. and Haeckel, M.: RV SONNE Fahrtbericht/cruise report SO239: EcoResponse assessing the ecology, connectivity and resilience of polymetallic nodule field systems, balboa (Panama)–Manzanillo (Mexico,) 11.03.-30.04. 2015, GEOMAR Reports, 2015.

Mewes, K., Mogollón, J. M., Picard, A., Rühlemann, C., Kuhn, T., Nöthen, K., and Kasten, S.: Impact of depositional and biogeochemical processes on small scale variations in nodule abundance in the Clarion-Clipperton Fracture Zone, Deep Sea Research Part I: Oceanographic Research Papers, 91, 125–141, 2014.

Oebius, H. U., Becker, H. J., Rolinski, S., and Jankowski, J. A.: Parametrization and evaluation of marine environmental impacts produced by deep-sea manganese nodule mining, Deep Sea Research Part II: Topical Studies in Oceanography, 48, 3453–3467, 2001.

Okazaki, M., Tsune, A., et al.: Exploration of Polymetallic Nodule Using AUV in the Central Equatorial Pacific, in: Tenth ISOPE Ocean Mining and Gas Hydrates Symposium, International Society of Offshore and Polar Engineers, 2013.

Ozturgut, E., Lavelle, J., Steffin, O., and Swift, S.: Environmental investigations during manganese nodule mining tests in the north equatorial Pacific in November 1978, 1980.

Park, S. H., Kim, D. H., Kim, C.-W., Park, C. Y., Kang, J. K., et al.: Estimation of coverage and size distribution of manganese nodules based on image processing techniques, in: Second ISOPE Ocean Mining Symposium, International Society of Offshore and Polar Engineers, 1997.

Pattan, J. and Kodagali, V.: Seabed topography and distribution of manganese nodules in the Central Indian Ocean, Mahasagar, 21, 7–12, 1988.

Peukert, A.: Correlation of ship- and AUV-based multibeam and side scan sonar analyses with visual AUV- and ROV-based data: Studies for Mn-nodule density quantification and mining-related environmental impact assessments, 2016.

Purser, A., Marcon, Y., Hoving, H.-J. T., Vecchione, M., Piatkowski, U., Eason, D., Bluhm, H., and Boetius, A.: Association of deep-sea incirrate octopods with manganese crusts and nodule fields in the Pacific Ocean, Current Biology, 26, R1268–R1269, 2016.

Rühlemann, C., Kuhn, T., Wiedicke, M., Kasten, S., Mewes, K., Picard, A., et al.: Current status of manganese nodule exploration in the German license area, in: Ninth ISOPE Ocean Mining Symposium, International Society of Offshore and Polar Engineers, 2011.

Sappington, J. M., Longshore, K. M., and Thompson, D. B.: Quantifying landscape ruggedness for animal habitat analysis: a case study using bighorn sheep in the Mojave Desert, Journal of wildlife management, 71, 1419–1426, 2007.

Schoening, T.: Results of nodule detection along AUV track SO239_19-1_AUV2 (Abyss_168) during SONNE cruise SO239, https://doi.org/10.1594/PANGAEA.879868, https://doi.org/10.1594/PANGAEA.879868, 2017a.

Schoening, T.: Results of nodule detection along AUV track SO239_28-1_AUV3 (Abyss_169) during SONNE cruise SO239, https://doi.org/10.1594/PANGAEA.879990, https://doi.org/10.1594/PANGAEA.879990, 2017b.

Schoening, T.: Source code for the Compact Morphology-based Nodule Delineation (CoMoNoD) algorithm, https://doi.org/10.1594/PANGAEA.875070, https://doi.org/10.1594/PANGAEA.875070, supplement to: Schoening, Timm; Jones, Daniel O B; Greinert, Jens (2017): Compact-Morphology-based poly-metallic Nodule Delineation. Scientific Reports, 7(1), https://doi.org/10.1038/s41598-017-13335-x, 2017c.

Schoening, T., Kuhn, T., and Nattkemper, T. W.: Estimation of poly-metallic nodule coverage in benthic images, in: Proc. of the 41st Conference of the Underwater Mining Institute (UMI), 2012.

Schoening, T., Kuhn, T., Jones, D. O., Simon-Lledo, E., and Nattkemper, T. W.: Fully automated image segmentation for benthic resource assessment of poly-metallic nodules, Methods in Oceanography, 15, 78–89, 2016.

Schoening, T., Jones, D. O., and Greinert, J.: Compact-morphology-based poly-metallic nodule delineation, Scientific Reports, 7, 13 338, 2017.

Sharma, R.: Indian Deep-sea Environment Experiment (INDEX):: An appraisal, Deep Sea Research Part II: Topical Studies in Oceanography, 48, 3295–3307, 2001.

Sharma, R.: Deep-sea mining: Economic, technical, technological, and environmental considerations for sustainable development, Marine Technology Society Journal, 45, 28–41, 2011.

Sharma, R. and Kodagali, V.: Influence of seabed topography on the distribution of manganese nodules and associated features in the Central Indian Basin: A study based on photographic observations, Marine geology, 110, 153–162, 1993.

Sharma, R., Nath, B. N., Parthiban, G., and Sankar, S. J.: Sediment redistribution during simulated benthic disturbance and its implications on deep seabed mining, Deep Sea Research Part II: Topical Studies in Oceanography, 48, 3363–3380, 2001.

Sharma, R., Sankar, S. J., Samanta, S., Sardar, A., and Gracious, D.: Image analysis of seafloor photographs for estimation of deep-sea minerals, Geo-marine letters, 30, 617–626, 2010.

Sharma, R., Khadge, N., and Jai Sankar, S.: Assessing the distribution and abundance of seabed minerals from seafloor photographic data in the Central Indian Ocean Basin, International journal of remote sensing, 34, 1691–1706, 2013.

Shirayama, Y. and Fukushima, T.: Responses of a meiobenthos community to rapid sedimentation, in: Proceedings, international symposium on environmental studies for deep-sea mining, 1997.

Skornyakova, N. and Murdmaa, I.: Local variations in distribution and composition of ferromanganese nodules in the Clarion-Clipperton Nodule Province, Marine Geology, 103, 381–405, 1992.

Thiel, H. and Tiefsee-Umweltschutz, F.: Evaluation of the environmental consequences of polymetallic nodule mining based on the results of the TUSCH Research Association, Deep Sea Research Part II: Topical Studies in Oceanography, 48, 3433–3452, 2001.

Trueblood, D. D., Ozturgut, E., et al.: The benthic impact experiment: A study of the ecological impacts of deep seabed mining on abyssal benthic communities, in: The Seventh International Offshore and Polar Engineering Conference, International Society of Offshore and Polar Engineers, 1997.

Tsune, A., Okazaki, M., et al.: Some Considerations about Image Analysis of Seafloor Photographs for Better Estimation of Parameters of Polymetallic Nodule Distribution, in: The Twenty-fourth International Ocean and Polar Engineering Conference, International Society of Offshore and Polar Engineers, 2014.

Vanreusel, A., Hilario, A., Ribeiro, P. A., Menot, L., and Arbizu, P. M.: Threatened by mining, polymetallic nodules are required to preserve abyssal epifauna, Scientific reports, 6, 26 808, 2016.

von Stackelberg, U. and Beiersdorf, H.: The formation of manganese nodules between the Clarion and Clipperton fracture zones southeast of Hawaii, Marine Geology, 98, 411–423, 1991.

Weiss, A.: Topographic position and landforms analysis, in: Poster presentation, ESRI user conference, San Diego, CA, vol. 200, 2001.

Wessel, P., Smith, W. H., Scharroo, R., Luis, J., and Wobbe, F.: Generic mapping tools: improved version released, Eos, Transactions American Geophysical Union, 94, 409–410, 2013.

Widmann, P., Kuhn, T., and Schulz, H.: Enrichment of mobilizable manganese in deep sea sediments in relation to Mn nodules abundance, in: EGU General Assembly Conference Abstracts, vol. 16, 2014.

Wilson, M. F., O'Connell, B., Brown, C., Guinan, J. C., and Grehan, A. J.: Multiscale terrain analysis of multibeam bathymetry data for habitat mapping on the continental slope, Marine Geodesy, 30, 3–35, 2007.

Wright, D., Pendleton, M., Boulware, J., Walbridge, S., Gerlt, B., Eslinger, D., Sampson, D., and Huntley, E.: ArcGIS Benthic Terrain Modeler (BTM), v. 3.0, Environmental Systems Research Institute, NOAA Coastal Services Center, Massachusetts Office of Coastal Zone Management, esriurl. com/5754, 2012.

5  Yamada, H., Yamazaki, T., et al.: Japan's ocean test of the nodule mining system, in: The Eighth International Offshore and Polar Engineering Conference, International Society of Offshore and Polar Engineers, 1998.

Yamazaki, T., Kajitani, Y., Barnett, B., Suzuki, T., et al.: Development of image analytical technique for resedimentation induced by nodule mining, in: Second ISOPE Ocean Mining Symposium, International Society of Offshore and Polar Engineers, 1997.

Yamazaki, T., Kajitani, Y., et al.: Deep-sea environment and impact experiment to it, in: The Ninth International Offshore and Polar Engi-
10  neering Conference, International Society of Offshore and Polar Engineers, 1999.

You, Z.-J.: The effect of suspended sediment concentration on the settling velocity of cohesive sediment in quiescent water, Ocean Engineering, 31, 1955–1965, 2004.

Zevenbergen, L. W. and Thorne, C. R.: Quantitative analysis of land surface topography, Earth surface processes and landforms, 12, 47–56, 1987.