# Peer review of "Understanding Mn-nodule distribution and evaluation of related deep-sea mining impacts using AUV-based hydroacoustic and optical data"

_Biogeosciences, 2017_

## Referee Comment (RC1) · R. Sharma (Referee) · 14 Dec 2017

Reviewer's comments for manuscript entitled 'Understanding Mn-nodule distribution and related deep-sea mining impacts using AUV-based hydroacoustic sensing and optical observations' Journal: Biogeosciences Authors: Peukert et al. A. General comments:

The manuscript deals with a detailed study on distribution of mn-nodules, associated bathymetry and impact of small-scale disturbance using ebibenthic sledge by high res-

olution mapping and imaging techniques. The study throws light on important issues such as distribution patterns of mn-nodules on the seafloor in a restricted area and the likely impact. The results offer sound inferences and important conclusions towards understanding the nodule occurrences with the associated environment that can be used as key inputs for planning of mining operations.

B. Specific comments:

In order to improve the readability and content of the manuscript, following suggestions may be considered :

1. Page 1 - Title: Suggest making it sharper. Delete 'understanding' and 'sensing' from 'Understanding Mn-nodule distribution and related deep-sea mining impacts using AUV-based hydroacoustic sensing and optical observations'.

2. Page 4 – Study area (line 28-29): Introduce a map showing general location of the study area with latitude, longitude and depth contours to give a general perspective to the reader (these details are not required for the subsequent figures given in the manuscript).

3. Page 8 – AUV based bathymetry ….abundance (line 7) : The word 'abundance' signifies 'quantity of resource per unit area (Kg/sqm)' where as here the nodule occurrence is expressed in 'percentage'. So 'abundance' should be replaced with 'coverage'.

4. Page 8 – Large-scale variability (line 24) : The term 'large-scale variability' is misleading and suggest that it can be replaced with 'Macrotopographic variability'.

5. Page 10 – Fig. 5D : Mean size of nodule is given as 6.7 cm2, 15cm2, 17.4 cm2. 'Mean size' should be replaced with 'Mean area' as size cannot be expressed as square.

6. Page 11 – Small-scale variability (line 21) : The term 'small-scale variability' is misleading and suggest that it can be replaced with 'Microtopographic variability'.

7. Page 17 – Broad-scale correlation. . ... (line 18) : Use of the term 'Broad scale . . .' is confusing and may be replaced with 'Regional scale. . ..' as it covers large area.

8. Page 17 - lines 25-28: Comment – Regional differences in nodule exposure (burial) could also be reason for this as nodules in Central Indian Ocean appear smaller due to more sediment cover as compared to those in the Pacific which could be due to differences in current velocities that influence settling of sediments.

9. Page 19 – Broad vs small scale : In cartography 'large (broad) scale' means representing small distances (area) and 'small scale' means covering larger distances (areas) for a given unit. To avoid any confusion for the reader, suggest that authors clarify the meaning of 'large scale' and 'small scale' or make necessary corrections (for example use the terms such as 'regional' and 'local').

10. Page 19-22 – Sediment plume resettling : This section is too long and without any breaks, so difficult to follow. Suggest that it could be divided into sub-sections with individual heading if possible and/or with paragraphs.

11. Page 22 - Conclusions – line 17 : Start new para from 'With respect to. . ..'

C. Technical / editorial comments:

1. Editorial corrections have been made in the document as track changes (attached separately). Authors are requested to see the same and accept / reject as suitable. 2. At a few places where it is not clear, a question (?) mark is inserted in the text where the authors can make necessary corrections / additions as required. 3. A few general editorial corrections required are as follows: i. Apply superscript for '2 (square)' wherever required ii. All references should be in bracket / parenthesis including author and year eg. (Page 2 – line 7 : Purser et al. 2016; Vanreusel et al. 2016). iii. Ship-based and AUV based may be replaced with ship-borne and AUV-borne

D. Recommendation:

It is recommended that the paper is suitable for publication after carrying out necessary

additions / corrections as suggested.

**Reviewer's comments for manuscript entitled 'Understanding Mn-nodule distribution and related deep-sea mining impacts using AUV-based hydroacoustic sensing and optical observations'**

Journal: Biogeosciences

Authors: Peukert et al.

**A. General comments:**

The manuscript deals with a detailed study on distribution of mn-nodules, associated bathymetry and impact of small-scale disturbance using ebibenthic sledge by high resolution mapping and imaging techniques. The study throws light on important issues such as distribution patterns of mn-nodules on the seafloor in a restricted area and the likely impact. The results offer sound inferences and important conclusions towards understanding the nodule occurrences with the associated environment that can be used as key inputs for planning of mining operations.

**B. Specific comments:**

In order to improve the readability and content of the manuscript, following suggestions may be considered :

1. Page 1 - Title: Suggest making it sharper. Eg. ' Mn-nodule distribution and related deep-sea mining impacts using AUV-based hydroacoustic  and optical observations'.

2. Page 4 – Study area (line 28-29): Introduce a map showing general location of the study area with latitude, longitude and depth contours to give a general perspective to the reader (these details are not required for the subsequent figures given in the manuscript).

3. Page 8 – AUV based bathymetry ....abundance (line 7) : The word 'abundance' signifies 'quantity of resource per unit area (Kg/sqm)' where as here the nodule occurrence is expressed in 'percentage'. So 'abundance' should be replaced with 'coverage'.

4. Page 8 – Large-scale variability (line 24) : The term 'large-scale variability' is misleading and suggest that it can be replaced with 'Macrotopographic variability'.

5. Page 10 – Fig. 5D : Mean size of nodule is given as 6.7 $cm^2$, 15$cm^2$, 17.4 $cm^2$. 'Mean size' should be replaced with 'Mean area' as size cannot be expressed as square.

**Fig. 1.** Reviewer's comments

[Figure]

**Understanding Mn-nodule distribution and related deep-sea mining impacts using AUV-based hydroacoustic sensing and optical observations**

Anne Peukert[1], Timm Schoening[1], Evangelos Alevizos[1], Kevin Köser[1], Tom Kwasnitschka[1], and Jens Greinert[1]

[1]GEOMAR Helmholtz-Center for Ocean Research Kiel, Wischhofstr. 1-3, 24148 Kiel, Germany

*Correspondence to:* Jens Greinert (jgreinert@geomar.de)

**Abstract.** In this study ship- and AUV-based multibeam data from the German Mn-nodule license area in the Clarion-Clipperton Zone (CCZ; eastern Pacific) are linked to ground truth data from optical imaging. Photographs obtained by an AUV enable semi-quantitative assessments of nodule coverage at a spatial resolution in the range of meters. Together with high resolution AUV bathymetry this revealed a correlation of small-scale terrain variations (<5m horizontally, <1m vertically) with nodule abundance. In the presented data set, increased nodule coverage could be correlated with slopes >1.8° and concave terrain. On a more regional scale, factors such as the geological setting (existence of horst and graben structures, sediment thickness, outcropping basement) and influence of bottom currents seem to play an essential role for the spatial variation of nodule abundance and the related hard substrate habitat. AUV imagery was also successfully employed to map the distribution of re-settled sediment following a disturbance and sediment cloud generation during a sampling deployment of an Epibenthic Sledge. Data from before and after the 'disturbance' allows a direct assessment of the impact. Automated image processing analyzed the nodule coverage at the seafloor, revealing nodule blanketing by resetting of suspended sediment within 16 hours after the disturbance. The visually detectable impact was spatially limited to a maximum of 100m distance from the disturbance track, downstream of the bottom water current. A correlation with high resolution AUV bathymetry reveals that the blanketing pattern varies in extent by tens of meters, strictly following the bathymetry, even in areas of only (?) slightly undulating seafloor (<1m vertical change). These results highlight the importance of detailed terrain knowledge when engaging in resource assessment studies for nodule abundance estimates and defining minable areas. At the same time, it shows the importance of high resolution mapping for detailed benthic habitat studies that show a heterogeneity at scales of 10m to 100m. Terrain knowledge is also needed to determine the scale of the impact by seafloor sediment blanketing during mining-operations.

*Copyright statement.* TEXT

**Fig. 2.** Manuscript with track changes

---

## Referee Comment (RC2) · T. Kuhn (Referee) · 8 Jan 2018

**Review of Ms No. Bg-2017-506**

Understanding Mn-nodule distribution and related deep-sea mining impacts using AUV-based hydroacoustic sensing and optical observations

By:

Anne Peukert et al.

*1. General Comments*

In the study presented in this manuscript AUV-based, high-resolution bathymetric data was linked with optical imaging. To my understanding, there are several objectives of this study:

(i) to get information of small-scale changes in seafloor morphology which is a pre-requisite for any future mining operation;

(ii) to get a semi-quantitative assessment of nodule coverage over an area which is larger than from normal TV-sled operations;

(iii) to combine both results in order to analyze the controlling factors of nodule coverage.

There is a second part of the study which deals with a "disturbance experiment" caused by an EBS station. This part of the study reports the extension of a sediment cloud based on the optical imaging of re-settled sediment as well as if there is a controlling influence of the small-scale bathymetry on the distribution of the re-settled sediments.

This is an interesting study mainly because of its small-scale and high-resolution character. The authors showed that there is no clear and simple correlation between any single parameter such as small-scale bathymetry and their derivates, near-bottom currents, sediment thickness and so on, on one side and the manganese nodule coverage/abundance on the other. When looking at the quality of the correlation between the different parameters it becomes rather clear that these correlation coefficients are very poor and can be called semi-quantitative at best. The reason for this poor correlation may be caused by the quality of the image analysis. For instance, Kuhn & Rathke (2017) showed in a study combining seafloor images and box corer data that the Mn nodule coverage (in %) is underestimated up to factor 5 based on image analyses compared to box corer data from the same area. They argue that the activity of both benthic fauna and near-bottom currents are the main reasons for this discrepancy. The authors of this manuscript should take this study by Kuhn & Rathke (2017) into account.

In the presented manuscript there is no information about precision and accuracy of the image analysis approach but this information is necessary and must be included. Moreover, Kuhn & Rathke (2017) showed in their study that a good correlation can be established between coverage data from box corer stations and image analysis for small-sized nodules ($R^2 = 0.645$; mean nodule size < 4cm diameter). However, only a weak and inverse correlation could be found for medium to large-sized nodules ($R^2 = 0.146$; mean nodule size > 4 cm). They showed that larger nodules are fewer in numbers and are covered by sediments to a larger degree. The authors of this study should present information on the mean nodule size and should provide information if they have found a similar correlation.

Despite the semi-quantitative character of the correlation between Mn nodule coverage and bathymetric data the authors show that nodule coverage may be below a certain threshold if the BPI50 > 0, slopes ≤ 1.8° and plan curvature values > -0.02 radians/m (page 11, line 8). A higher coverage instead occurs at steeper slopes (> 1.5°), negative BPI50 values (-200 to 0) and negative plan curvature (< -0.02) interpreted as morphological depressions.

Even if I doubt the absolute number of 12.5 % nodule coverage as the threshold value I still think that the above-mentioned semi-quantitative correlations between nodule coverage and bathymetric parameters are true and important findings of this study. However, more real ground truth data from box corer stations would be necessary to verify the threshold value.

I also wonder if there are any correlations between AUV-based backscatter data (such as BS intensity) and nodule coverage?

Another approach would be to analyze the nodule coverage and the hydroacoustic data based on artificial neural networks or on random forest. Did the authors try these approaches?

The second part of the study is not known from other studies with respect to the small-scale and high resolution (at least to my knowledge). The fact that small-scale morphologic changes play a significant role in controlling the re-settlement of sediments is very interesting and important, even if this correlation also is only semi-quantitative.

What the manuscript generally lacks is real ground-truth data for Mn nodule coverage which can only be obtained from sampling with box corers. Is there any such information from the working area, e.g. from other cruises? I know that the BGR has carried out several expeditions to this area within their exploration campaign.

**2. Specific Comments**

**Abstract**

First sentence: Optical imaging data are no real ground truth data. If they could be linked with nodule coverage/abundance from box corer stations of this area, then one could speak of "ground-truth data". Otherwise, the authors should change this sentence removing the word "ground-truth".

**Nodule coverage vs. nodule abundance**

The authors sometimes use the term "nodule abundance" and sometimes "nodule coverage". There is a significant difference between both: abundance means the mass of nodules per area (e.g., in kg/m²) and coverage means the seafloor areal fraction covered by nodules in %. From image analysis only the coverage can be detected and this is what the authors mean in their manuscript (e.g. refer to Fig. 5). Therefore, the authors should only use the term "nodule coverage" in the manuscript.

With respect to the precision and accuracy of the nodule coverage detected from the image analysis I can state the following: The BGR took four box corers more or less parallel and a few hundred meters to the north of the line shown in Fig. 5A. The mean nodule size areas

measured on all nodules from these box corers are: 23 cm², 23 cm², 24 cm² and 20 cm² from west to east. So, the mean nodule size is rather constant in this area and is by factor up to 3.4 higher when measured on real nodules from box corers compared to data from images. To my understanding this discrepancy is the main reason for the poor correlation coefficients and it may be caused by the observation that nodules are covered by sediments to a variable degree. But in images only the part of the nodules not covered by sediments can be analyzed and this may lead to a significant underestimation in both coverage and size of nodules as we can see it in the data presented in this manuscript.

Pit structures

The occurrence of pit structures may not only be restricted to larger depressions as stated on page 8, line 12, but could also be controlled by E-W trending linear structures as Fig. 4C may suggest.

A pit structure was sampled during SO140 with a box corer (station 107KG). There were no nodules on the sediment surface but two nodule layers at greater sediment depth (16 and 32 cm below surface; Kuhn et al., 2015). This contradicts the interpretation of the authors of this study of how larger nodules in the pit structure could have formed (page 18, line 20/21). BGR data suggest that larger nodules have a larger diagenetic fraction and thus should have grown faster. A larger diagenetic fraction, however, is only possible at sites with higher sedimentation rates and/or higher TOC content. A slightly higher sedimentation rate in areas of higher nodule coverage is also discussed by the authors of this study further down in the manuscript (page 18, line 30/31). Moreover, the pit structures are interpreted as sites of higher sedimentation rates (page 18, lines 35ff.). Why should other depressional sites behave differently in terms of the sedimentation regime?

At sites with stronger bottom currents, e.g. at sites where the near-bottom currents are channelized, nodules do have a higher hydrogenetic content and they are generally smaller and occur in higher numbers (BGR data, e.g. Rühlemann et al., 2012).

The discussion on the pit structures on page 19, lines 1-13 is wrong. During cruise SO240 one such pit was sampled with box corer and gravity corer. Pore water chemistry was not different from other sampling sites outside the pits (Kuhn et al., 2015). Moreover, heat flow measurements over such pit structures did not show any temperature anomalies. Therefore, the most likely interpretation of these structures is a type of marine carst caused by a widespread seawater circulation in the basaltic crust underneath the sediments. At sites where faults in the basaltic crust reach into the upper carbonate-bearing sediments seawater which is undersaturated in $CO_3$ moves into and reacts with the sediments causing dissolution of carbonates which in turn leads to the pit-like collapse structures on the seafloor (Kuhn et al., 2017).

Small-scale bathymetry and nodule coverage

Page 8, line 26-27: Figure 4b indicates that there is a steeper slope in sub-area A2 whereas this area is characterized by higher nodule coverage compared to sub-area A1 (Fig. 5B). This is contradictory to the statement given at page 8, line 26-27. Even if in this part of the manuscript ship-based bathymetry is compared with image analysis one should not make a

statement which is already proven uncorrect by other data (here the AUV-based high-res. bathymetry data). I suggest to omit this sentence and to discuss the different results in the discussion section.

Again the correlations provided in Fig. 6 between nodule coverage and several other parameters are very weak to non-existing, except for the median nodule size and the BPI50 in sub-area A2.

The interpretation of the distribution of the nodule coverage presented in Fig. 7 is based on these weak correlations. How does the predicted low coverage from Fig. 7 correspond with the coverage data from the AUV photo survey? Please provide a scatter plot with nodule coverage from image analyses (x-axis) and nodule coverage from the combination of hydro-acoustic data (y-axis).

Sediment plume settling

Page 13, line 30: How was the threshold of 8% nodule coverage as complete blanketing defined? Why not 0%?

Discussion about particles size in a sediment cloud (page 20/21): The assumption of Stoke's law to describe the sinking behavior of the plume particles is incorrect. Flocculation occurs at large-scale as experimental and modelling results from the JPIO project "Mining Impacts" have shown (pers. comm. A. Vink). Thus, the particles sizes should be much larger than 29 µm on average and the sinking velocities should be rather between 0.5 and 3 m/s. These higher sinking velocities may require a plume height greater than 1.6 m…?

Mn nodule growth (page 18-19)

The work of von Stackelberg & Beiersdorf (1991) describes the influence of different parameters on the Mn nodule growth. This work should be taken into account by the authors.

The publication of Knobloch et al. (2017) indicates the importance of the bathymetry and the conditions on the seafloor for Mn nodule formation.

The citation of Mewes et al. (2014) on page 18/19 may not be correct. Mewes et al. (2014) describe that at sites with medium to large-sized nodules a smaller percentage of clay particles have been found in the surface sediments. This may be due to increased activity of near-bottom currents which has removed part of the clay particles. The remaining sediment may have contained a relatively higher proportion of mobilizable Mn which was then available for Mn nodule formation.

*3. Technical Corrections*

Mixing of abundance and coverage throughout the manuscript. Please correct – see above.

Always use the term "ferromanganese nodules" in the text starting with a small letter except at the beginning of sentences.

Pay attention to the correct statement of references, e.g., always use parenthesis within a sentence (cf. page 2, line 7 and at many other lines in the text).

Page 1, line 18: mining operations (no -).

Page 2, line 21: 12 km²

Page 6, line 8/30: data citation is missing

Page 14, line 7: it must read East instead of West

Page 15, line 3: it must read west-facing slope

 Page 21, line 36-39: Something is wrong with the grammar

Table A1: AUV MB (Fig. 4, not Fig. 2)

Table A2: What is the difference between mineable ridges and ridges, flat depression and depression, mineable deep depression and deep depression?

Table A3: How are the different classes (mineable versus un-mineable) defined?

Table A6: Why is BPI440st used in this table and not BPI50st?

Page 29, 1st reference: year is missing.

*4. Cited references*

Knobloch, A., Kuhn, T., Rühlemann, C., Hertweg, T., Zeissler, K.-O., Noack, S. 2017. Predictive mapping of the nodule abundance and mineral resource estimation in the Clarion-Clipperton Zone using artificial neural networks and classical geostatistical methods. In: R. Sharma (Ed.): Deep-Sea Mining: Resource Potential, Technical and Environmental Considerations. *Springer International*, Cham, pp. 189 – 212.

Kuhn, T., Rathke, M. (2017). Report on visual data acquisition in the field and interpretation for SMnN. Deliverable D1.31 of the EU-Project *Blue Mining*. www.bluemining.eu/downloads. BGR Hannover, 34 pp.

Kuhn, T., Versteegh, G.J.M., Villinger, H., Dohrmann, I., Heller, C., Koschinsky, A., Kaul, N., Ritter, S., Wegorzewski, A.V., Kasten, S., 2017c. Widespread seawater circulation in 18–22 Ma oceanic crust: Impact on heat flow and sediment geochemistry. Geology, 45/9: 799-802. https://doi.org/10.1130/G39091.1

Rühlemann, C. and Shipboard Scientific Party (2012). Biodiversity, Geology and Geochemistry of the German and French License Areas for the Exploration of polymetallic Nodules in the Equatorial NE Pacific: BIONOD Cruise Report, Volume 1 German License Area. Hannover, 109 pp.

Von Stackelberg, U., and Beiersdorf, H., 1991, The formation of manganese nodules between the Clarion and Clipperton Fracture-Zones southeast of Hawaii: Marine Geology, v. 98, no. 2-4, p. 411-423.

---

## Referee Comment (RC3) · J. Kim (Referee) · 12 Jan 2018

Review comment of Ms# BG-2017-506

"Understanding Mn-nodule distribution and related deep-sea mining impacts using AUV-based hydroacoustic sensing and optical observations" by Anne Peukert et al.

General comments

The paper presents new results for variation of nodule distribution in seafloor and for dispersion of sediment plume by disturbance experiment. The main contents of the manuscript, i.e. (1) nodule distribution in accordance with changes in seafloor morphology and (2) evaluation of environmental impact experiment, were conducted by several previous studies in 90's and early 2000's. However, this paper provides valuable and important data by application of high resolution mapping and imaging analyses using AUV survey.

Such small-scale and high resolution work will be essential for preparing the development of deep-sea mining and establishing relevant environmental guidelines. And thus, the paper will be of interest to scientists and engineers of various fields related to deep sea minerals. The most part I think the authors provide sound interpretation and conclusion supported by their data and analyses. However, I found some part, mainly about nodule coverage and abundance, need to be expressed more clearly (see the specific comments below). I think the paper needs improvement mostly in use of the term "nodule coverage" for more restricted meaning. Once the authors have done this I think the paper will be suitable for publication.

Specific comments (including technical comments)

Section 1.1 Pg. 2, lines 17-24. As authors mentioned, detailed small-scale investigations are rare in previous work. However, advantages of the small-scale investigation are not described well in the manuscript. It will be helpful if authors can provide some specific issues on nodule distribution which cannot be understood in previous conventional ship-based studies in the Introduction.

Pg. 2 line 8 the reference should be corrected

Pg. 2 line 21 use superscript for km2

Section 1.3. Pg. 5. Fig. 2. Geographic Information (i.e. latitude and longitude) needs to be added in the figures showing study area. It will be more helpful if the authors can provide an index map which shows location of study area with some useful information (regional topography or sediment type, for example)

Pg. 6 line 7 and line 30 add the references for data sources

Section 3.1 Nodule coverage

Variation of nodule coverage associated with seafloor morphology is one of main contexts of the manuscript. Sometimes the term nodule coverage is used as nodule abundance in sections of results and discussion, which makes some confusion. For example, what is the meaning of variation of nodule coverage? Does it mean the difference of actual abundance of nodule or difference of occurrence of nodule (i.e. variation of sediment cover). In other words, the results indicate a small (or local) scale variation of nodule coverage. Does it caused by difference of nodule growth or just reflecting different sediment distribution without significant change of condition for nodule growth? To avoid such confusion, I recommend that authors at least provide a definition for the term "coverage" used in this study possibly in "Introduction" or "Methodology".

As the authors mentioned in the manuscript, the photographs cannot reflect accurate nodule abundance. If this is the case, the authors should examine their interpretation on the observed variation of nodule coverage carefully. For example, the authors wrote that "The presented data show that favorable nodule growth/occurrence conditions coincide with gentle sloping sites and low relief depressions, where sediment is assumed to accumulate slowly"(Pg. 18 line 24-25). However, if the variation of coverage cannot represent the actual change of abundance, we cannot say that a certain location of slightly high coverage is more favorable site for nodule growth. In my view, natural variation of sediment resuspension by bottom current in accordance with topographical change appears as more plausible explanation for the observed variation of nodule coverage.

Of course, the authors should have some freedom of interpretation, but some of interpretation appears to be speculation without supporting data. Thus, I recommend the authors only use the term "nodule coverage", provide a definition or meaning of variation of nodule coverage in this study, and reorganize the manuscript accordingly.

Pg. 8 line 26-27, Fig. 5. The description in the sentence is not clearly shown in Fig. 5C. When variation of nodule coverage is shown together with the bathymetric profile in Figure 5C, it will be easy to see the correlation. Please add color indexing layer above the bathymetric profile in Fig. 5C.

Pg. 13 line 12 and 14. Please check the figure number.

Pg. 16 Fig. 10. Providing large photos of same location before and after the EBS will be helpful. This can be added in Fig. 10 or be presented as appendix figure.

Pg. 16 line 5. I cannot understand the meaning of size of area, 0.49 km2. Does it an area of photo survey in Abyss 168 or Abyss 169 in Fig. 9? If so, please add information.

Pg. 17 line 6. What is the CoMoNoD? Need explanation or information for readers who are not aware of the algorithm by Schoening (2017).

Pg. 17 line 30 check the misspelling "and"

Pg. 19 line 13 Water currents can be replaced by Bottom currents

Pg. 19. Some of paragraphs are too long and need splitting. This is especially for the last section of discussion (4.3 Sediment plume re-settling), but also for some other part of the manuscript.

Please use parenthesis for reference citation within a sentence.

---

## Author Response (AR1)

**Reviewer 1**

**Reviewer 1**: Page 1 - Title: Suggest making it sharper. Delete 'understanding' and 'sensing' from 'Understanding Mn-nodule distribution and related deep-sea mining impacts using AUV-based hydroacoustic sensing and optical observations'.
**Authors Comment**: Title has been changed (DC).
**Document Changes:** Understanding Mn-nodule distribution and evaluation of related deep-sea mining impacts using AUV-based hydroacoustic and optical data.

**R1:** Page 4 – Study area (line 28-29): Introduce a map showing general location of the study area with latitude, longitude and depth contours to give a general perspective to the reader (these details are not required for the subsequent figures given in the manuscript).
**AC:** We added the coordinates for the center of the working area in Figure caption Figure 2.
**DC:** Black squares mark the study area (center 117°1 W 11°51N) shown in Figure 3.

**R1:** Page 8 – AUV based bathymetry . . ..abundance (line 7) : The word 'abundance' signifies 'quantity of resource per unit area (Kg/sqm)' whereas here the nodule occurrence is expressed in 'percentage'. So 'abundance' should be replaced with 'coverage'.
**AC:** Has been corrected.

**R1:** Page 8 – Large-scale variability (line 24) : The term 'large-scale variability' is misleading and suggest that it can be replaced with 'Macrotopographic variability'.
**AC:** It has been replaced (DC) to make this clearer.
**DC:** Broad scale variability (less detailed, correlation with ship-based bathymetric data, resolution 100-1000m)

**R1:** Page 10 – Fig. 5D : Mean size of nodule is given as 6.7 cm2, 15cm2, 17.4 cm2. 'Mean size' should be replaced with 'Mean area' as size cannot be expressed as square.
**AC:** To our understanding $cm^2$ can be used as an expression for size and we consider it as more suitable here than the expression "Mean-area"; of course, we are aware of buried areas of the nodules.

**R1:** Page 11 – Small-scale variability (line 21) : The term 'small-scale variability' is misleading and suggest that it can be replaced with 'Microtopographic variability'.
**AC**: Has been corrected (DC).
**DC:** Local scale variability (more detailed, correlation with AUV-based bathymetric data, resolution 1-100m)

**R1:** Page 17 – Broad-scale correlation. . ... (line 18) : Use of the term 'Broad scale . . .' is confusing and may be replaced with 'Regional scale. . ..' as it covers large area.
**AC:** Has been replaced as suggested.

**R1:** Page 17 - lines 25-28: Comment – Regional differences in nodule exposure (burial) could also be reason for this as nodules in Central Indian Ocean appear smaller due to more sediment cover as compared to those in the Pacific which could be due to differences in current velocities that influence settling of sediments.

**AC:** This should indeed be mentioned here and has been added to the text p.24, edited manuscript.

**DC:** Varying considerations of scale and regional differences in nodule exposure between different oceans across different studies have thus led to partly contradicting statements of the relationship between the Mn-nodule coverage/size and bathymetric settings.

**R1:** Page 19 – Broad vs small scale : In cartography 'large (broad) scale' means representing small distances (area) and 'small scale' means covering larger distances (areas) for a given unit. To avoid any confusion for the reader, suggest that authors clarify the meaning of 'large scale' and 'small scale' or make necessary corrections (for example use the terms such as 'regional' and 'local').

**AC:** This remark has been taken into account. We changed the passages through the text to avoid any confusion.

**R1:** Page 19-22 – Sediment plume resettling : This section is too long and without any breaks, so difficult to follow. Suggest that it could be divided into sub-sections with individual heading if possible and/or with paragraphs.

**AC:** Paragraphs and sub-sections have been added in the section (see manuscript, section 4.3).

**R1:** Page 22 - Conclusions – line 17: Start new para from 'With respect to. . ..'

**AC:** Has been changed as suggested.

**Technical / editorial comments:**

**R1:** At a few places where it is not clear, a question (?) mark is inserted in the text where the authors can make necessary corrections / additions as required.

**AC:** Thank you for the remarks. We made a number of additions/corrections at those places (see manuscript, changes are tracked).

**AC:** p.5, line 4-5: In these lines only the Experiments and year of conduction are named, not any references. MMAJ: BIE conducted in 1997 within the area of the Marcus-Wake Seamounts in the North Pacific Ocean. Reference: Yamada and Yamazaki, 1998;

**R1:** A few general editorial corrections required are as follows:

i. Apply superscript for '2 (square)' wherever required

**AC:** Has been corrected.

**R1:** ii. All references should be in bracket / parenthesis including author and year eg. (Page 2 – line 7 : Purser et al. 2016; Vanreusel et al. 2016).

**AC:** The citing format is one accepted format of the journal. But since it was commented from all other reviewers as well it has been changed to the suggested format.

**R1:** iii. Shipbased and AUV based may be replaced with ship-borne and AUV-borne

**AC:** We would like to stick with AUV-based and ship-based as this is a typical way to indicate with which platform the data have been acquired.

Cited References:

Yamada, H., and T. Yamazaki. 1998. "Japan's Ocean Test of the Nodule Mining System." 1998/1/1/.

**Reviewer 2**

**1 General Comments**

**Reviewer 2:** The authors of this study should present information on the mean nodule size …
**Authors Comment:** Considering the potential inaccuracy of nodule detection and separation of the image analysis tool, the application of quantiles on the size distribution allows a more accurate interpretation of the data (Peukert, 2016). Therefore, a mean size value would not be appropriate here. An explanation was added to the Appendix (see DC).

**Document Changes:** Interpretation of Mn-nodule size results

Considering the probable error in correctly detecting nodules by the image-analyzing tool, the application of quantiles of the size distribution allows a better interpretation of the data. It is suggested not to use size values of the smallest and largest 1 % of the quantile calculation, due to the above mentioned error source. The graph in figure A1 illustrates the quantiles of the calculated sizes of two images, which clearly differ from each other. The graph correctly displays a size difference between both images, indicating larger nodules for image #29302. This shows that the tool can be reasonably applied to calculate the nodule size. The best differentiation however exists for the 50% - 75% quantile. Towards larger and smaller size values the curves approach each other which points towards the detection of similar – non nodule - features in both images. Therefore, the median values are considered to best represent nodule size differences between images/areas. Since truly correct nodule identification by the tool cannot be ensured for this quantile, size values should not be seen and used as absolute values, but rather indicators of changes between areas that are compared.

[Figure]

[Figure]

**Figure A1:** Two example images which clearly differ in nodule size and coverage. Graph A shows the calculated quantiles of Mn-nodule sizes in two example images (B, C).The results indicate that the strongest difference can be seen between the 50 and 75 % quantile.

**R2:** Moreover, Kuhn & Rathke (2017) showed in their study that a good correlation can be established between coverage data from box corer stations and image analysis for small-sized nodules […].The authors of this study […] should provide information if they have found a similar correlation. […]In the presented manuscript there is no information about precision and accuracy of the image analysis approach but this information is necessary and must be included.

**AC:** Indeed it would have been nice to correlate box-core data with image data. Unfortunately, we do not have corresponding image and box-core information. To do this properly, the seafloor photographs from before the sampling would be required, exactly knowing where the box-corer will take the sample. Alternatively, the AUV could have made a photo survey before and after the sampling. We do not have such data and thus cannot accommodate the request of the reviewer.

**R2:** […] Even if I doubt the absolute number of 12.5 % nodule coverage as the threshold value…

**AC:** The 12.5% Mn nodule coverage is of course not an absolute value, as it has been discussed several times. The calculated coverage results are in a range between 7-24%. However, the majority of 99% is between 8 and 17 %. Values below and above are outliers and can be considered as inaccuracy of the automatic nodule detection by "unusual" objects in the image (like tracks or a fish for example). 12.5% was set as a threshold, since it is the median and mean of this majority and is the highest occurring coverage amount (Figure 1, this comment section). Furthermore, applying this threshold, the difference in Mn nodule coverage follows bathymetric structures (especially in A2); of course, the differences in Mn nodule coverage are very low (and probably not relevant for resource assessment, which is not the goal of this study) but so are the morphological undulations within the studied area. To make this clearer, an explanation of the threshold value was added to the text (see DC, p.12, line 17ff.).

**DC:** Based on the automated image analyses, the majority of the seafloor shows nodule coverage values between 8% and 17% (Figure 5A). Values below and above this range (<1%) are to neglect, since they are caused by "unusual" objects (like tracks or organisms) in the images. In the following examinations the threshold between 'low' and 'high' Mn-nodule coverage is set at 12.5%; which is the analyzed mean coverage value of the considered range. In the eastern A2 sub-area a greater proportion of higher coverage values (13-16%) can be observed.

[Figure]

*Figure 1: Frequency of Mn-nodule coverage results of the entire AUV photo survey Abyss 168. Mean value 13%.*

**R2:** However, more real ground truth data from box corer stations would be necessary to verify the threshold value.

**AC:** That is true and would be part of further investigations, as mentioned above. The following sentence has been changed to mention this: section 4.1. p.24, line 4-5

**DC:** For absolute accurate resource assessments and verification of the results, detailed sampling based on this study would need to follow.

**R2:** I also wonder if there are any correlations between AUV-based backscatter data (such as BS intensity) and nodule coverage?

**AC:** Unfortunately, the BS data of the area analyzed in this study were not usable due to technical errors. The data from other areas though look very promising. These are part of other studies, which are currently in preparation.

**R2:** Another approach would be to analyze the nodule coverage and the hydroacoustic data based on artificial neural networks or on random forest. Did the authors try these approaches?

**AC:** This approach is part of other studies and was not pursued here. As mentioned above, the data used for this study were "data of opportunity" and not acquired to perform statistically correct machine learning approaches aiming at extrapolation of nodule coverage / resource assessments.

**R2:** What the manuscript generally lacks is real ground-truth data for Mn nodule coverage which can only be obtained from sampling with box corers. Is there any such information from the working area, e.g. from other cruises? I know that the BGR has carried out several expeditions to this area within their exploration campaign.

**AC:** As already mentioned above, box core sampling would be the next step based on these results for verification. This would require highly detailed sampling at exactly the same area analyzed here. In the publication of Kuhn et al. (2016), the box core stations are too far away (at least 500m) and also the BGR BC stations are located within this area, are too far away (Figure 2, this reviewer section). Thus, this data cannot be used as ground truth validation of our results. However, two

tracks of visual observations, which were also carried out from BGR, match more or less the AUV photo track of this area and provide similar observations (Peukert, 2016).

[Figure]

*Figure 2: Box core stations marked as colored dots conducted by BGR within the study area.*

**2. Specific Comments**

Abstract

**R2:** First sentence: Optical imaging data are no real ground truth data. If they could be linked with nodule coverage/abundance from box corer stations of this area, then one could speak of "ground-truth data". Otherwise, the authors should change this sentence removing the word "ground-truth".
**AC:** "Ground-truth" here means the visually from the AUV images detectable nodule coverage on the sediment surface, not the absolute coverage including the buried nodules. The term "ground-truth" seems reasonable to us for this study.

Nodule coverage vs. nodule abundance

**R2:** The authors sometimes use the term "nodule abundance" and sometimes "nodule coverage". There is a significant difference between both: abundance means the mass of nodules per area (e.g., in kg/m²) and coverage means the seafloor areal fraction covered by nodules in %. From image analysis only the coverage can be detected and this is what the authors mean in their manuscript (e.g. refer to Fig. 5). Therefore, the authors should only use the term "nodule coverage" in the manuscript.
**AC:** Thank you for this remark. We changed this throughout the text.

**R2:** […] To my understanding this discrepancy is the main reason for the poor correlation coefficients and it may be caused by the observation that nodules are covered by sediments to a variable degree. But in images only the part of the nodules not covered by sediments can be analyzed and this may lead to a significant underestimation in both coverage and size of nodules as we can see it in the data presented in this manuscript.
**AC:** We are aware of the discrepancy between the visually detectable sediment surface and the "real" nodule coverage on the seafloor which is discussed in the paper (e.g. p.17, line 2ff. or p. 18,

lines 27-29). The results of Kuhn & Rathke (2017) regarding the accuracy is taken into account and mentioned in the text (section 4.1, p.24, l.4), see DC). This study aimed to show possible correlations between morphology on different scales and small-scale relative (not absolute!) changes in nodule coverage; such small scale-changes of course require very detailed sampling which needs to be taken out to verify the results as a next step. However, as mentioned in the paper, for habitat mapping purposes, nodules are considered as a hard substrate habitat where only the unburied part of the nodules on the sediment surface is relevant, making the visual mapping technique a very useful tool (p. 2, line 8-9, p. 17, line 4-6).

**DC:** Photographs only provide information of the sediment surface and thus will not be able to detect buried/sediment-covered Mn-nodules (Sharma and Kodagali, 1993; Sharma et al., 2010, 2013), resulting in an underestimation of the absolute Mn nodule coverage (Kuhn and Rathke, 2017).

Pit Structures

**R2:** The occurrence of pit structures may not only be restricted to larger depressions as stated on page 8, line 12, but could also be controlled by E-W trending linear structures as Fig. 4C may suggest.
**AC:** Yes, they could be controlled by E-W trending linear structures, however the AUV-mapped area does not allow the statement that E-W structures are more important than 'negative BPI' in general.

**R2:** A pit structure was sampled during SO140 with a box corer (station 107KG). There were no nodules on the sediment surface but two nodule layers at greater sediment depth (16 and 32 cm below surface; Kuhn et al., 2015). This contradicts the interpretation of the authors of this study of how larger nodules in the pit structure could have formed (page 18, line 20/21). [...]
**AC:** We believe the reviewer considered larger structures in the SO140 cruise than the pits in focus of this study: "sizes from several tens of meters to 150m in diameter with a maximum depth of 4m" (p.8,line 11). However, at this point the reviewer mixes this study's "depressions" which are proposed to contain higher nodule coverages with this study's "pits", where no nodules can be seen at the sediment surface. We argue that nodules could be buried here (p.27, line 4-5, see edited manuscript), which is in agreement with your findings.

**R2:** [...]BGR data suggest that larger nodules have a larger diagenetic fraction and thus should have grown faster. A larger diagenetic fraction, however, is only possible at sites with higher sedimentation rates and/or higher TOC content. A slightly higher sedimentation rate in areas of higher nodule coverage is also discussed by the authors of this study further down in the manuscript (page 18, line 30/31). Moreover, the pit structures are interpreted as sites of higher sedimentation rates (page 18, lines 35ff.). Why should other depressional sites behave differently in terms of the sedimentation regime?
**AC:** We distinguish between "depressions" and "pits", which occur within wider depressions (section 3.1, p. 12, line 9-11). In section 4.2.2. where the lack of Mn nodule coverages in these structures are discussed, we changed the text so the difference becomes more prominent (see DC).
**DC:** p. 26, line 29ff (see edited manuscript): Rather special for the presented data set are the pronounced pit structures, observed throughout the AUV-mapped area with very little to no Mn-nodules observed at the sediment surface, which is in contradiction to the wider depressions, where higher Mn-nodule coverage was observed.

**AC:** Our interpretation is that only in the pits the sedimentation rate is too high for nodules to appear at the sediment surface (because they were buried, p.18, lines 32-34), which is in agreement with the

reviewer's above mentioned findings. The pits are likely to be younger structures; nodules have formed within the depressions first. The collapse forming the pits occurred later. The nodules within the pits were then buried by sediments following the gravity to the deepest point and accumulating there.

We added the interpretation of pits being marine karsts and the associated reference to the interpretations (see DC).
**DC:** p. 27, lines 5-7 (edited manuscript): The formation process of the pits is unclear, but could be karst structures, as proposed by Kuhn et al. (2017), which are younger than the Mn nodule formation and which would point towards Mn-nodule burial within the pits.

**R2:** At sites with stronger bottom currents, e.g. at sites where the near-bottom currents are channelized, nodules do have a higher hydrogenetic content and they are generally smaller and occur in higher numbers (BGR data, e.g. Rühlemann et al., 2012).
**AC:** What kind of morphological changes would be needed to increase bottom currents, what would be the size of the morphological change? Do we talk about kilometer-, 100m- or meter-scale? This 'scaling issue' makes these results hard to compare with previous studies, which dealt with coarser scales, than this one.

**R2:** The discussion on the pit structures on page 19, lines 1-13 is wrong. During cruise SO240 one such pit was sampled with box corer and gravity corer. Pore water chemistry was not different from other sampling sites outside the pits (Kuhn et al., 2015). Moreover, heat flow measurements over such pit structures did not show any temperature anomalies. […]
**AC:** As mentioned above, we believe the reviewer did not sample a structure in a comparable size in the mentioned cruise. Moreover, it is hard to precisely sample exactly within the pits of such size, especially if the high resolution bathymetry is not available.

Small-scale bathymetry and nodule coverage

**R2:** Figure 4b indicates that there is a steeper slope in sub-area A2 whereas this area is characterized by higher nodule coverage compared to sub-area A1 (Fig. 5B). This is contradictory to the statement given at page 8, line 26-27.
**AC:** The trouble is that that ship- and AUV-obtained bathymetry show different correlations. Therefore, it is not possible to apply one statement to different scales. This is one of the main findings of this study and is discussed in section 4.2.4. p.19.

**R2:** The interpretation of the distribution of the nodule coverage presented in Fig. 7 is based on these weak correlations. How does the predicted low coverage from Fig. 7 correspond with the coverage data from the AUV photo survey? Please provide a scatter plot with nodule coverage from image analyses (x-axis) and nodule coverage from the combination of hydro-acoustic data (y-axis).
**AC:** This links to the Machine Learning approach, which was not done in this study.

Sediment plume settling

**R2:** Page 13, line 30: How was the threshold of 8% nodule coverage as complete blanketing defined? Why not 0%?
**AC:** 8% was the minimum value, because the algorithm sometimes misinterprets shadows as nodules (no area with 0% nodule coverage).

**R2:** Discussion about particles size in a sediment cloud (page 20/21): The assumption of Stoke's law to describe the sinking behavior of the plume particles is incorrect. Flocculation occurs at large-scale as experimental and modelling results from the JPIO project "Mining Impacts" have shown (pers. comm. A. Vink).

**AC:** This is written in the text (p.20, line 30/31). Flocculation could also lead to increased friction lowering the sinking velocities, as discussed in p. 20, line 31

**R2:** Thus, the particles sizes should be much larger than 29 μm on average and the sinking velocities should be rather between 0.5 and 3 m/s. These higher sinking velocities may require a plume height greater than 1.6 m…?

**AC:** 29 μm is the median particle size in the area, disregarding flocculation (p.20, line 26-27). This scenario and the simple application of Stokes Law was just used to highlight the difficulty in estimating the distribution of a mining-induced sediment cloud, since several factors, such as flocculation / aggregation have to be taken into account and it is hard to make a statement on how such massive sediment plumes will behave in a real mining scenario and what difference these factors make. Nevertheless, the calculated plume height created by the EBS in the experiment is approximately 0.96 to 1.6m (p.20, line 30) in agreement with measured ADCP data (p.20, lines 32-35).

Regarding "sinking velocities should be rather between 0.5 and 3 m/s": How did the reviewer get these values?

Mn nodule growth (page 18-19)

**R2:** The work of von Stackelberg & Beiersdorf (1991) describes the influence of different parameters on the Mn nodule growth. This work should be taken into account by the authors.

**AC:** The mentioned work has been taken into account and was cited (p. 17, line 20).

**R2:** The citation of Mewes et al. (2014) on page 18/19 may not be correct. Mewes et al. (2014) describe that at sites with medium to large-sized nodules a smaller percentage of clay particles have been found in the surface sediments. This may be due to increased activity of near-bottom currents which has removed part of the clay particles. The remaining sediment may have contained a relatively higher proportion of mobilizable Mn which was then available for Mn nodule formation.

**AC:** "[…]higher sedimentation rate in a low current regime would also mean a higher accumulation of clay size particles, which are proposed to hinder nodule growth Mewes et al. (2014)." To our understanding this means the same in reversion? However, "hinder" has been changed to "not favorable for" (p.29, line1).

**3. Technical Corrections**

**R2:** Mixing of abundance and coverage throughout the manuscript. Please correct – see above.
**AC:** Has been corrected.

**R2:** Always use the term "ferromanganese nodules" in the text starting with a small letter except at the beginning of sentences.
**AC:** Mn-nodule was introduced as an abbreviation for ferromanganese nodule in p.2 line 5. It was changed from a capital letter to starting with a small letter, as suggested.

**R2:** Pay attention to the correct statement of references, e.g., always use parenthesis within a sentence (cf. page 2, line 7 and at many other lines in the text).
**AC:** Has been changed.

**R2:** Page 1, line 18: mining operations (no -).
**AC:** Has been corrected.

**R2:** Page 2, line 21: 12 km$^2$
**AC:** Has been corrected.

**R2:** Page 6, line 8/30: data citation is missing
**AC:** Has been added.

**R2:** Page 14, line 7: it must read East instead of West
**AC:** It is correct as it is. Three different things are named here.

**R2**: Page 15, line 3: it must read west-facing slope
AC: No, the purple shadings indicate east-facing slopes.

**R2:** Page 21, line 36-39: Something is wrong with the grammar
**AC:** These lines are not present? Do you mean another page?

**R2:** Table A1: AUV MB (Fig. 4, not Fig. 2)
**AC:** Has been corrected.

**R2:** Table A2: What is the difference between mineable ridges and ridges, flat depression and depression, mineable deep depression and deep depression?
Table A3: How are the different classes (mineable versus un-mineable) defined?
**AC:** Thank you for the remark. The following explanation was added to the Figure captions:
**DC**: p.6, line 3 and p.9, line3: The terms "minable" and "unminable" are defined by slope threshold ("minable": slope <= 3°; "unminable": slope >3°).

**R2:** Table A6: Why is BPI440st used in this table and not BPI50st?
**AC:** BPI440st seemed more reasonable for an overview description of the AUV-mapped area, which is why it was used for Fig. 4. BPI50st was used for the small-scale analysis because this BPI-scale detects the single pit structures. Table A6 summarizes the statistics for the descriptive derivatives of the AUV-mapped area, displayed in Figure 4.

**R2:** Page 29, 1st reference: year is missing.
**AC:** Reference year is not missing in our document?

**Cited References**

Kuhn, T., Rathke, M. (2017). Report on visual data acquisition in the field and interpretation for SMnN. Deliverable D1.31 of the EU-Project *Blue Mining*. www.bluemining.eu/downloads. BGR Hannover, 34 pp.

Peukert, Anne. 2016. "Correlation of ship- and AUV-based multibeam and side scan sonar analyses with visual AUV- and ROV-based data: Studies for Mn-nodule density quantification and

mining-related environmental impact assessments." unpublished MSc. Thesis MSc. Thesis, Institute of Geosciences, Christian-Albrechts-Universität zu Kiel.

**Reviewer 3**

**Reviewer 3**: Section 1.1 Pg. 2, lines 17-24. As authors mentioned, detailed small-scale investigations are rare in previous work. However, advantages of the small-scale investigation are not described well in the manuscript. It will be helpful if authors can provide some specific issues on nodule distribution which cannot be understood in previous conventional ship-based studies in the Introduction.

**Authors Comment:** As shown in this study, there are local scale changes in Mn-nodule occurrence, and this variability can be correlated to detailed morphological changes, which is relevant information to understand Mn-nodule forming – processes (p.2, line 18-19).

The local variability studied here is of importance for habitat-studies (section 4.1 p.16, line 5ff) as next to the topographic setting, the nodule availability is of importance for determining local marine habitats i.e. hard grounds. Following your suggestion this has been added in section 1 (p.2, line 18, see DC).

**Document Changes:** Moreover, the substrate changes considered in this study provide relevant information for estimating size and heterogeneity of local-scale habitats.

**AC:** Last but not least, only data in the detailed scale considered here can provide reliable information on morphology, which is essential for planning possible mining tracks, since not the entire terrain is suitable and obstacles need to be taken into account for the development of mining gear. This information was added in section 5 p.34, lines 27-29 (see DC).

**DC:** Areas that appeared suitable of mining (slopes <=3°) in ship-based bathymetric data showed steeper relief (slopes >3°) in higher resolution AUV-based data.

**R3:** Pg. 2 line 8 the reference should be corrected
**AC:** Has been corrected.

**R3:** Pg. 2 line 21 use superscript for km2
**AC:** Has been corrected.

**R3:** Section 1.3. Pg. 5. Fig. 2. Geographic Information (i.e. latitude and longitude) needs to be added in the figures showing study area. It will be more helpful if the authors can provide an index map which shows location of study area with some useful information (regional topography or sediment type, for example).
**AC:** Since the study area is located within the German claim area for resource exploration the exact location is not published within this study (this was discussed with the BGR as contractor of the area). We added the coordinates of the center of the working area in Figure 2 caption (DC). The regional topography is shown in Figures 2 and 3 and is described in the text (section 1.3).
**DC:** Black squares mark the study area (center 117°1 W 11°51N) shown in Figure 3.

**R3:** Pg. 6 line 7 and line 30 add the references for data sources
**AC:** Has been added.

**R3:** Section 3.1 Nodule coverage:
[…] Thus, I recommend the authors only use the term "nodule coverage", provide a definition or meaning of variation of nodule coverage in this study, and reorganize the manuscript accordingly.

**AC:** In section 2 p.8, line 31 the Mn nodule coverage considered here is defined (percent coverage per image). Following your suggestion "per image" was added for clarification. We mistakenly used the term "abundance" and changed it to the correct term "coverage" throughout the manuscript.
**DC:** p.8, line 31: **"**percent coverage per image"

**R3:** Pg. 8 line 26-27, Fig. 5. The description in the sentence is not clearly shown in Fig. 5C. When variation of nodule coverage is shown together with the bathymetric profile in Figure 5C, it will be easy to see the correlation. Please add color indexing layer above the bathymetric profile in Fig. 5C.
**AC:** We considered your suggestion and edited Figure 5. The bathymetric profile with the resolution of the AUV-based bathymetry, color coded with the Mn-nodule coverage, was added to the ship-based bathymetric profile.

**R3:** Pg. 13 line 12 and 14. Please check the figure number.
**AC:** Has been corrected.
**DC:** Figure 6B

**R3:** Pg. 16 Fig. 10. Providing large photos of same location before and after the EBS will be helpful. This can be added in Fig. 10 or be presented as appendix figure.
**AC:** Considering the navigational error coming along with the AUV data (discussed in p.9, line1), it is not possible to show one photo of the exact same location before and after the EBS experiment even though the exact track was programmed for the AUV survey (p.6, lines 29-30). However, it is reasonable to compare the entire track, where specific patterns (containing of various continuous images) can be used for the recognition of the same areas (p.9, lines 2-3). Due to the absence of large features on the seafloor in the studied area (large enough to not being buried by the resettling sediments) making a recognition of the exact same area possible, such a comparison figure is not shown in this study. We believe the mosaic in Figure 10 shows the burial effect of the EBS-induced sediment cloud sufficiently.

**R3:** Pg. 16 line 5. I cannot understand the meaning of size of area, 0.49 km2. Does it an area of photo survey in Abyss 168 or Abyss 169 in Fig. 9? If so, please add information.
**AC:** it means the total area that was covered with AUV imagery. For clarification this information was added as suggested to the mentioned passage.
**DC**: "0.49km$^2$ that is completely photo-mapped"

**R3:** Pg. 17 line 6. What is the CoMoNoD? Need explanation or information for readers who are not aware of the algorithm by Schoening (2017).
**AC:** It is briefly described in section 2 p.6 line 32ff. For further information the reference is cited and the content of the paper does not need to be given here.

**R3:** Pg. 17 line 30 check the misspelling "and"
**AC:** Has been corrected.

**R3:** Pg. 19 line 13 Water currents can be replaced by Bottom currents
**AC:** Has been replaced as suggested.

**R3:** Pg. 19. Some of paragraphs are too long and need splitting. This is especially for the last section of discussion (4.3 Sediment plume re-settling), but also for some other part of the manuscript.
**AC:** Sub-sections have been added in section 4.3.

**R3:** Please use parenthesis for reference citation within a sentence.

**AC:** Has been corrected.

**R3:** Please use parenthesis for reference citation within a sentence.

**AC:** Has been corrected.

[revised manuscript text omitted]